# VISTA: Visual-Textual Knowledge Graph Representation Learning

Jaejun Lee    Chanyoung Chung    Hochang Lee
Sungho Jo    Joyce Jiyoung Whang*
School of Computing, KAIST
{jjlee98, chanyoung.chung, hochang.lee91, shjo, jjwhang}@kaist.ac.kr

## Abstract

Knowledge graphs represent human knowledge using triplets composed of entities and relations. While most existing knowledge graph embedding methods only consider the structure of a knowledge graph, a few recently proposed multimodal methods utilize images or text descriptions of entities in a knowledge graph. In this paper, we propose visual-textual knowledge graphs (VTKGs), where not only entities but also triplets can be explained using images, and both entities and relations can accompany text descriptions. By compiling visually expressible commonsense knowledge, we construct new benchmark datasets where triplets themselves are explained by images, and the meanings of entities and relations are described using text. We propose VISTA, a knowledge graph representation learning method for VTKGs, which incorporates the visual and textual representations of entities and relations using entity encoding, relation encoding, and triplet decoding transformers. Experiments show that VISTA outperforms state-of-the-art knowledge graph completion methods in real-world VTKGs.

## 1 Introduction

Knowledge graphs provide diverse human knowledge in a structured form, representing each fact as a triplet consisting of entities and a relation. Knowledge graph representation learning methods (Ji et al., 2022) aim to convert the entities and relations into a set of representation vectors which can be utilized in predicting missing triplets or in other applications such as commonsense reasoning (Lin et al., 2019) and question answering models (Liu et al., 2020). While most existing knowledge graph embedding methods focus solely on the structure of a knowledge graph to learn the representations (Lacroix et al., 2018; Sun et al., 2019), using additional images or text descriptions can result in much better representations. A few

―――――――――
*Corresponding author

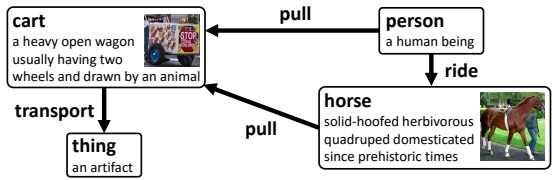

(a) Existing Multimodal Knowledge Graph

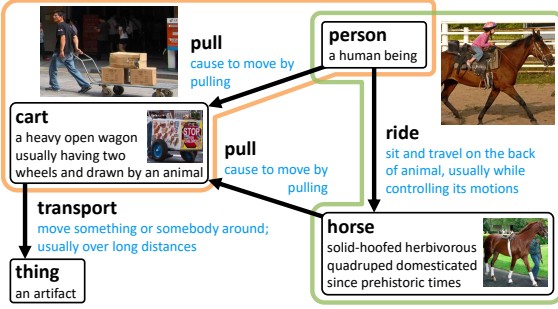

(b) Visual-Textual Knowledge Graph (VTKG)

Figure 1: While existing methods assume only entities have their images and text descriptions, our VTKG additionally considers cases where a triplet itself is represented by an image, and relations have text descriptions.

recently proposed multimodal knowledge graph completion methods (Zhao et al., 2022; Li et al., 2023) consider the case where entities can have their images and text descriptions; Figure 1 (a) is a form of knowledge graph existing multimodal knowledge graph completion methods consider.

We focus on the fact that some triplets can be more intuitively expressed by their images. For example, in Figure 1 (b), ⟨person, ride, horse⟩ and ⟨person, pull, cart⟩ have images, where the triplets themselves are represented by images, providing visual insights. On the other hand, existing knowledge graphs do not provide images for triplets and may lack visually expressible triplets because they are constructed by extracting information mainly from texts. Therefore, we propose forming a knowledge graph by extracting information from images and utilizing those images to represent visual commonsense knowledge.

To this end, we compile visual phrases in triplets and their images from diverse computer vision tasks, including visual relationship detection (Lu et al., 2016), human-object interaction detection (Chao et al., 2018), and visual knowledge extraction (Sadeghi et al., 2015). We propose Visual-Textual Knowledge Graphs (VTKGs) where the entities and triplets can be represented by images, and the entities and relations have their text descriptions as shown in Figure 1 (b). We build new VTKG benchmark datasets containing visually expressible commonsense knowledge, the entities' and triplets' images, and detailed descriptions of the entities and relations.

To learn representations of entities and relations in VTKGs, we propose the **VIS**ual-**T**extu**A**l (VISTA) knowledge graph representation learning method that utilizes not only the structure of a visual-textual knowledge graph but also the visual and textual features extracted from images and text descriptions. In particular, when a triplet is provided with an image, VISTA learns the visual representation of the relation in the given triplet. The resulting visual representation of the relation is also utilized when the relation appears in other triplets, which enhances the entire representation learning process. To the best of our knowledge, VISTA is the first knowledge graph representation learning method that learns the visual representations for visually expressible relations. We design VISTA by proposing three transformers: entity encoding, relation encoding, and triplet decoding transformers. The entity and relation encoding transformers represent entities and relations using their visual and textual feature vectors, whereas the triplet decoding transformer predicts a missing entity in a triplet using a masking scheme. Experimental results on four real-world datasets demonstrate that VISTA outperforms 10 different state-of-the-art knowledge graph completion methods. Our datasets and codes are available at `https://github.com/bdi-lab/VISTA`.

## 2 Related Work

**Visual Commonsense Reasoning** There have been some attempts to extract visual knowledge from images by using visually verifiable relations (Sadeghi et al., 2015; Chen et al., 2013). However, they were not studied in the context of multimodal knowledge graph representation learning, and some datasets are not accessible now. More

recently, the Visual Genome dataset (Krishna et al., 2017) has been released, where diverse computer vision datasets are merged. However, this dataset contains heterogeneous information which is not in the form of triplets or is hard to be considered as commonsense knowledge. On the other hand, our VTKG datasets provide visual commonsense knowledge in the form of triplets with images and can seamlessly enlarge existing knowledge bases. We believe our work can be utilized in visual commonsense reasoning (Zellers et al., 2019) and visual question answering (VQA) (Antol et al., 2015).

**Knowledge Integration** Ilievski et al. have attempted to examine the characteristics of information from different sources by manually categorizing the kind of knowledge (Ilievski et al., 2021). From a multimodal learning point of view, adding different modalities to existing knowledge bases has been considered (Zhu et al., 2022), e.g., adding images or texts to entities in a knowledge graph. Different from these approaches, our VTKGs are proposed to represent visually expressible knowledge, and our benchmark datasets are created by a fine-level alignment of entities and relations from different sources using WordNet synsets (Miller, 1995). Details are described in Section 3.2.

**Multimodal Knowledge Graph Completion** While some knowledge graph embedding methods utilize images of entities (Xie et al., 2017; Wang et al., 2021; Oñoro-Rubio et al., 2019; Liu et al., 2019), some recently proposed multimodal methods consider both images and text descriptions of entities (Pezeshkpour et al., 2018; Wang et al., 2019). For example, MoSE (Zhao et al., 2022) and IMF (Li et al., 2023) learn modality-specific representations and make predictions using the representations from different modalities. Also, OTKGE (Cao et al., 2022) proposes an optimal transport to align multi-modal embeddings, while MKGformer (Chen et al., 2022) conducts multi-level fusion using a hybrid transformer. Unlike VISTA, all these existing methods assume that only entities can have images or descriptions and do not consider the cases where an image represents a triplet itself or relations have descriptions.

## 3 Visual-Textual Knowledge Graphs

We describe how we collect visual commonsense knowledge and create VTKG benchmark datasets.

| | $|\mathcal{V}|$ | $|\mathcal{R}|$ | $|\mathcal{T}|$ | $|\mathcal{I}_{\text{ent}}|$ | $|\mathcal{I}_{\text{tri}}|$ | $|\mathcal{D}_{\text{ent}}|$ | $|\mathcal{D}_{\text{rel}}|$ | VTKG-I | VTKG-C |
|---|---|---|---|---|---|---|---|---|---|
| VRD | 130 | 88 | 842 | 14,945 | 10,372 | 129 | 80 | ✓ | ✓ |
| UnRel | 31 | 19 | 44 | 1,636 | 939 | 31 | 18 | ✓ | ✓ |
| HICO-DET | 96 | 157 | 516 | 232,830 | 129,936 | 92 | 155 | ✓ | ✓ |
| VisKE | 605 | 563 | 2,422 | 0 | 0 | 591 | 552 | | ✓ |
| ConceptNetW | 6,260 | 2,421 | 14,919 | 0 | 0 | 5,922 | 2,205 | | ✓ |
| WN18RR++ | 41,105 | 11 | 93,003 | 70,349 | 0 | 41,105 | 0 | | ✓ |

Table 1: The original sources of our two VTKG datasets. The first four rows are datasets representing visual commonsense knowledge, and the last two rows are knowledge bases.

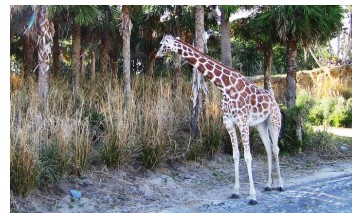

⟨giraffe, eat, grass⟩

Figure 2: Visual Relationship Detection. Given an image, a short visual phrase is created using a triplet.

## 3.1 Representing Visual Commonsense Knowledge as Triplets and Images

There can be many different design choices for extracting information from images, depending on the type of information we want to extract. Among others, we look into computer vision tasks returning a visual phrase in the form of ⟨object$_1$, predicate, object$_2$⟩. We assemble visual commonsense knowledge using four benchmark datasets from three different computer vision tasks: VRD (Lu et al., 2016) and UnRel (Peyre et al., 2017) for visual relationship detection, HICO-DET (Chao et al., 2018) for human-object interaction detection, and VisKE (Sadeghi et al., 2015) for visual knowledge extraction. All these tasks return a short visual phrase in the triplet format to describe a given image. For example, Figure 2 shows an output of visual relationship detection. Among the four datasets, VRD, UnRel, and HICO-DET provide the triplets with images, whereas VisKE only provides triplets but not images. When collecting the visual phrases from these datasets, we filter out image-specific phrases that are hard to be considered as commonsense knowledge, e.g., ⟨dog, on_the_right_of, vase⟩.

## 3.2 Defining and Creating VTKGs

We define a VTKG as $G = (\mathcal{V}, \mathcal{R}, \mathcal{T}, \mathcal{I}, \mathcal{D})$ where $\mathcal{V}$ is a set of entities, $\mathcal{R}$ is a set of relations, $\mathcal{T}$ is a set of triplets, $\mathcal{I}$ is a set of images, and $\mathcal{D}$ is a set of text descriptions. Since an image can be attached to an entity or a triplet, $\mathcal{I} = \mathcal{I}_{\text{ent}} \cup \mathcal{I}_{\text{tri}}$ where $\mathcal{I}_{\text{ent}}$ indicates a set of images attached to entities and $\mathcal{I}_{\text{tri}}$ is a set of images attached to triplets. Also, $\mathcal{D} = \mathcal{D}_{\text{ent}} \cup \mathcal{D}_{\text{rel}}$ where $\mathcal{D}_{\text{ent}}$ and $\mathcal{D}_{\text{rel}}$ indicate the descriptions of entities and relations, respectively. Throughout the paper, we use script upper cases for sets, boldfaced upper cases for matrices, and boldfaced lower cases for vectors.

To create VTKG datasets, we use well-known knowledge bases, ConceptNet (Speer et al., 2017) and WN18RR (Dettmers et al., 2018). We note that some tail entities in ConceptNet include predicates. For example, ⟨spider, CapableOf, kill_a_fly⟩ should be converted to ⟨spider, kill, fly⟩ to be consistent with other triplets in the other datasets; this consistency is important because we combine triplets from different sources to create our VTKG datasets where the same entities and relations should be appropriately aligned across different datasets. We manually inspected the triplets in ConceptNet for three weeks, resulting in the **ConceptNetW** dataset, which is a subset of ConceptNet, containing 14,919 triplets. On the other hand, in the original WN18RR, the entity types are missing. Thus, some entities are not mapped into a unique synset ID which is defined in WordNet (Miller, 1995), resulting in incorrectly representing different entities as one entity. To fix this, we manually mapped those entities to one of the five types: noun, verb, adjective, adjective satellite, and adverb. The resulting dataset is **WN18RR++**.

Using the visual commonsense knowledge datasets and knowledge bases, we create two VTKG datasets shown in Table 1. In **VTKG-I**, we combine visual commonsense datasets where all triplets have their images. In **VTKG-C**, we merge all visual commonsense datasets and two knowledge bases, ConceptNetW and WN18RR++.

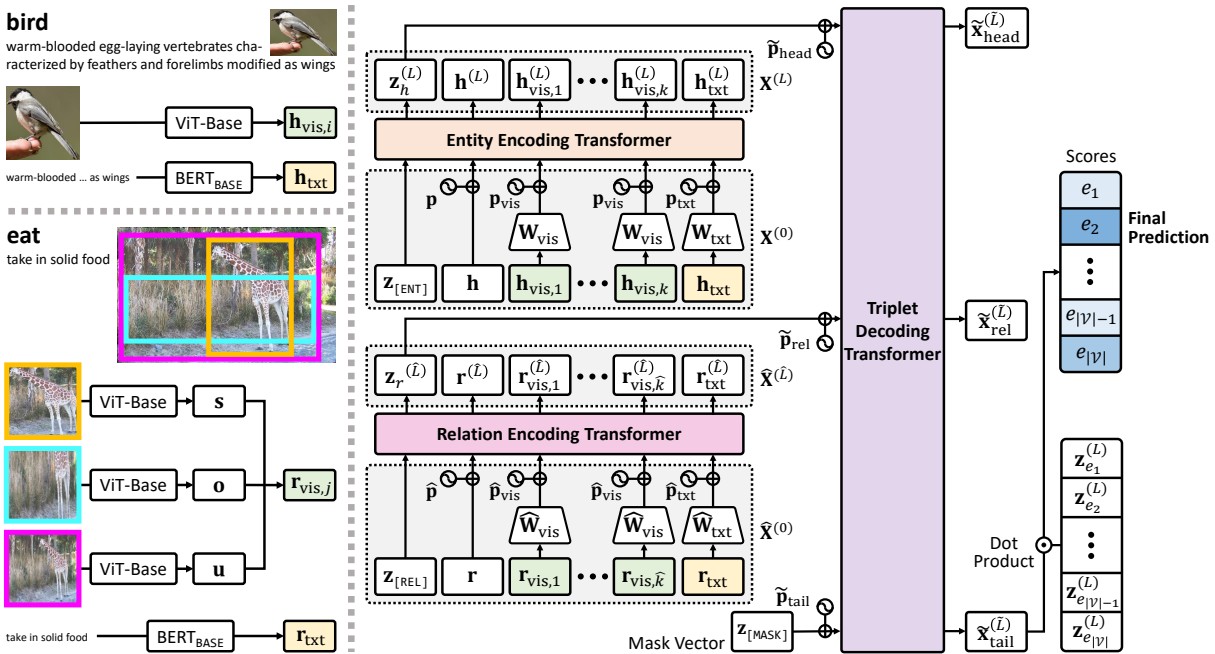

Figure 3: Overview of VISTA. We extract visual and textual features using ViT-Base and BERT$_{\text{BASE}}$. The entity and relation encoding transformers compute the entity and relation representations, respectively. The resulting representations are fed to the triplet decoding transformer, which predicts a missing entity in a triplet.

Constructing VTKG datasets requires much more effort than simply unioning the triplets because the triplets from different sources are usually written using different vocabularies, and the semantics of the same word should be determined depending on the context. For example, the same concept can be written using different words, e.g., human or person. Also, the same word can indicate different meanings depending on the context, e.g., ⟨diamond, scratch, glass⟩ or ⟨bartender, fill, glass⟩. To refine the expressions, we mapped each entity and relation into a synset provided in Word-Net. This mapping requires us to select the most appropriate synset of each word in a triplet by examining the meaning of each synset and the semantics of the word in context. **We replaced the entity and the relation with the corresponding synset ID so that all triplets are rephrased using the synsets defined in WordNet**. In this way, we consolidated the vocabularies used in VRD, UnRel, HICO-DET, VisKE, and ConceptNetW. Details about this process are described in Appendix A. Since WN18RR++ is a subset of WordNet, it did not require this post-processing.

## 4 VISTA: Learning Representations in Visual-Textual Knowledge Graphs

We propose VISTA shown in Figure 3. We extract visual and textual features from images and text de-

scriptions using pretrained ViT (Dosovitskiy et al., 2021) and BERT (Devlin et al., 2019). By utilizing the extracted features, VISTA learns representations of entities and relations, which are computed by the entity and relation encoding transformers, respectively. Based on the resulting entity and relation representations, VISTA predicts a missing entity in a triplet using the triplet decoding transformer.

### 4.1 Extracting Visual and Textual Features

In VTKGs, images can be associated with an entity or a triplet. To extract visual features from the images, we employ ViT pretrained on ImageNet-21k (Deng et al., 2009). We resize an input of ViT-Base as $224 \times 224$ and consider the final hidden state corresponding to [class] token as the visual feature vector. If an entity has multiple images, the entity can have multiple visual feature vectors, one from each image. In practice, we select the top $k$ images per entity by computing pairwise dot product similarities of the visual feature vectors.

When an image is given for a triplet $\langle s, r, o \rangle$, we first extract the visual features of entities, $s$ and $o$, by cropping bounding boxes, each of which contains $s$ and $o$ and feeding each bounding box to ViT-Base. Let $\mathbf{s}$ and $\mathbf{o}$ denote the resulting vectors. To consider the visual attribute of relation $r$, we define the minimal union bounding box that con-

tains both entities $s$ and $o$ and compute the visual feature vector $\mathbf{u}$ by feeding this bounding box to ViT-Base. Then, we define $\mathbf{r}_{\text{vis}} = [\mathbf{s}; \mathbf{o}; \mathbf{u}]$ where ; is a vertical concatenation, $\mathbf{s}, \mathbf{o}, \mathbf{u} \in \mathbb{R}^{d_{\text{vis}}}$, and $d_{\text{vis}} = 768$. The visual representation vector of $r$ is computed by learning a projection matrix acting on $\mathbf{r}_{\text{vis}}$. Details are described in Section 4.3. For each relation, we select the top $\hat{k}$ images based on the pairwise dot product similarities of the $\mathbf{r}_{\text{vis}}$ values.

To extract textual features from descriptions, we employ the pretrained BERT. For each given text description of an entity or a relation, we first tokenize the description, feed the tokenized sequence to BERT$_{\text{BASE}}$, and use the final hidden state corresponding to [CLS] token as the textual feature vector. For entities and relations without descriptions, we feed their labels in natural language to BERT$_{\text{BASE}}$. The dimension of a textual feature vector is set to $d_{\text{txt}} = 768$.

## 4.2 Entity Encoding Transformer

Let us consider an entity $h$ with $k$ visual features, $\mathbf{h}_{\text{vis},1}, \cdots, \mathbf{h}_{\text{vis},k} \in \mathbb{R}^{d_{\text{vis}}}$, and the textual feature $\mathbf{h}_{\text{txt}} \in \mathbb{R}^{d_{\text{txt}}}$. The input of the entity encoding transformer is defined by

$$\mathbf{X}^{(0)} = \Big[\mathbf{z}_{[\text{ENT}]}\|(\mathbf{h} + \mathbf{p})\|(\mathbf{W}_{\text{vis}}\mathbf{h}_{\text{vis},1} + \mathbf{p}_{\text{vis}})\| \cdots$$
$$\|(\mathbf{W}_{\text{vis}}\mathbf{h}_{\text{vis},k} + \mathbf{p}_{\text{vis}})\|(\mathbf{W}_{\text{txt}}\mathbf{h}_{\text{txt}} + \mathbf{p}_{\text{txt}})\Big]$$
$$= \Big[\mathbf{z}_h^{(0)}\|\mathbf{h}^{(0)}\|\mathbf{h}_{\text{vis},1}^{(0)}\| \cdots \|\mathbf{h}_{\text{vis},k}^{(0)}\|\mathbf{h}_{\text{txt}}^{(0)}\Big]$$

where $\mathbf{z}_{[\text{ENT}]} \in \mathbb{R}^d$ is a learnable vector that is shared across all entities, $d$ is the dimension of a representation vector, $\mathbf{h} \in \mathbb{R}^d$ is a learnable vector for $h$, $\mathbf{W}_{\text{vis}} \in \mathbb{R}^{d \times d_{\text{vis}}}$ and $\mathbf{W}_{\text{txt}} \in \mathbb{R}^{d \times d_{\text{txt}}}$ are learnable projection matrices for converting the visual and textual features to their representation vectors, respectively, $\mathbf{p}, \mathbf{p}_{\text{vis}}, \mathbf{p}_{\text{txt}} \in \mathbb{R}^d$ are the positional encodings of the learnable vector, visual representation vectors, and the textual representation vector, respectively, and $\|$ denotes a horizontal concatenation. After $L$ transformer encoder layers (Vaswani et al., 2017), we get the final representation matrix

$$\mathbf{X}^{(L)} = \Big[\mathbf{z}_h^{(L)}\|\mathbf{h}^{(L)}\|\mathbf{h}_{\text{vis},1}^{(L)}\| \cdots \|\mathbf{h}_{\text{vis},k}^{(L)}\|\mathbf{h}_{\text{txt}}^{(L)}\Big]$$

and we use $\mathbf{z}_h^{(L)}$ as the representation vector of the entity $h$.

## 4.3 Relation Encoding Transformer

Given a relation $r$, consider $\mathbf{r}_{\text{vis},1}, \cdots, \mathbf{r}_{\text{vis},\hat{k}} \in \mathbb{R}^{3d_{\text{vis}}}$ described in Section 4.1. Let us consider images containing the relation $r$. Specifically, consider the $j$-th image containing a triplet $\langle s_j, r, o_j \rangle$. Recall that $\mathbf{r}_{\text{vis},j}$ is created by concatenating the visual features of $s_j, o_j$, and the union bounding box containing $s_j$ and $o_j$. We compute the initial visual representation vector of $r$ for the $j$-th image by $\widehat{\mathbf{W}}_{\text{vis}}\mathbf{r}_{\text{vis},j}$ where a learnable projection matrix $\widehat{\mathbf{W}}_{\text{vis}}$ is introduced. A visual representation vector of $r$ is learned per image while sharing the projection matrix. The way we compute the visual representation of $r$ can be considered as a generalization of UVTransE (Hung et al., 2021) since $\widehat{\mathbf{W}}_{\text{vis}}$ allows flexible operations between the feature vectors of $s$, $o$, and their union bounding box, whereas UVTransE employs a fixed constraint. On the other hand, let $\widehat{\mathbf{W}}_{\text{txt}} \in \mathbb{R}^{d \times d_{\text{txt}}}$ be a learnable matrix for converting the textual feature vector $\mathbf{r}_{\text{txt}}$ of $r$ to its textual representation vector.

The input of the relation encoding transformer is

$$\widehat{\mathbf{X}}^{(0)} = \Big[\mathbf{z}_{[\text{REL}]}\|(\mathbf{r} + \widehat{\mathbf{p}})\|(\widehat{\mathbf{W}}_{\text{vis}}\mathbf{r}_{\text{vis},1} + \widehat{\mathbf{p}}_{\text{vis}})\| \cdots$$
$$\|(\widehat{\mathbf{W}}_{\text{vis}}\mathbf{r}_{\text{vis},\hat{k}} + \widehat{\mathbf{p}}_{\text{vis}})\|(\widehat{\mathbf{W}}_{\text{txt}}\mathbf{r}_{\text{txt}} + \widehat{\mathbf{p}}_{\text{txt}})\Big]$$
$$= \Big[\mathbf{z}_r^{(0)}\|\mathbf{r}^{(0)}\|\mathbf{r}_{\text{vis},1}^{(0)}\| \cdots \|\mathbf{r}_{\text{vis},\hat{k}}^{(0)}\|\mathbf{r}_{\text{txt}}^{(0)}\Big]$$

where $\mathbf{z}_{[\text{REL}]} \in \mathbb{R}^d$ is a learnable vector that is shared across all relations, $\mathbf{r} \in \mathbb{R}^d$ is a learnable vector for $r$, $\widehat{\mathbf{p}}, \widehat{\mathbf{p}}_{\text{vis}}, \widehat{\mathbf{p}}_{\text{txt}} \in \mathbb{R}^d$ are the positional encodings of the learnable vector, visual representation vectors, and the textual representation vector, respectively. After $\widehat{L}$ transformer encoder layers, we get the final representation matrix

$$\widehat{\mathbf{X}}^{(\widehat{L})} = \Big[\mathbf{z}_r^{(\widehat{L})}\|\mathbf{r}^{(\widehat{L})}\|\mathbf{r}_{\text{vis},1}^{(\widehat{L})}\| \cdots \|\mathbf{r}_{\text{vis},\hat{k}}^{(\widehat{L})}\|\mathbf{r}_{\text{txt}}^{(\widehat{L})}\Big]$$

and we use $\mathbf{z}_r^{(\widehat{L})}$ as the representation vector of the relation $r$.

## 4.4 Triplet Decoding Transformer

Given a triplet $\langle h, r, ? \rangle$, the triplet decoding transformer predicts the missing entity using $\mathbf{z}_h^{(L)}$, $\mathbf{z}_r^{(\widehat{L})}$, and a learnable mask vector $\mathbf{z}_{[\text{MASK}]} \in \mathbb{R}^d$. The input of the triplet decoding transformer is

$$\widetilde{\mathbf{X}}^{(0)} = [(\mathbf{z}_h^{(L)} + \widetilde{\mathbf{p}}_{\text{head}})\|(\mathbf{z}_r^{(\widehat{L})} + \widetilde{\mathbf{p}}_{\text{rel}})\|(\mathbf{z}_{[\text{MASK}]} + \widetilde{\mathbf{p}}_{\text{tail}})]$$
$$= [\widetilde{\mathbf{x}}_{\text{head}}^{(0)}\|\widetilde{\mathbf{x}}_{\text{rel}}^{(0)}\|\widetilde{\mathbf{x}}_{\text{tail}}^{(0)}]$$

where $\widetilde{\mathbf{p}}_{\text{head}}, \widetilde{\mathbf{p}}_{\text{rel}}, \widetilde{\mathbf{p}}_{\text{tail}} \in \mathbb{R}^d$ are the positional encodings of the head entity, relation, and tail entity, respectively. After applying $\widetilde{L}$ transformer encoder layers, we have

$$\widetilde{\mathbf{X}}^{(\widetilde{L})} = \left[ \widetilde{\mathbf{x}}_{\text{head}}^{(\widetilde{L})} \| \widetilde{\mathbf{x}}_{\text{rel}}^{(\widetilde{L})} \| \widetilde{\mathbf{x}}_{\text{tail}}^{(\widetilde{L})} \right]$$

and calculate the score $y_j$ of an entity $e_j$ by $y_j = \widetilde{\mathbf{x}}_{\text{tail}}^{(\widetilde{L})} \cdot \mathbf{z}_{e_j}^{(L)}$ where $\mathbf{z}_{e_j}^{(L)}$ is the representation vector of $e_j$ computed from the entity encoding transformer. We predict the missing entity to be the entity with the highest score. Similarly, we can also make a prediction for $\langle ?, r, t \rangle$. We use the cross entropy loss in our implementation.

# 5 Experiments

We evaluate the performance of VISTA using four real-world datasets by comparing it to 10 different knowledge graph completion methods. We implement VISTA using PyTorch (Paszke et al., 2019) with the Adam optimizer (Kingma and Ba, 2015). We apply dropout (Srivastava et al., 2014) and use the cosine annealing scheduler (Loshchilov and Hutter, 2017).

## 5.1 Datasets and Experimental Setup

We use four datasets shown in Table 2; VTKG-I and VTKG-C are real-world VTKG datasets introduced in Section 3.2. While WN18 (Bordes et al., 2013) and FB15K237 (Toutanova and Chen, 2015) are benchmark datasets used in other multimodal knowledge graph completion research (Zhao et al., 2022; Chen et al., 2022), WN18 has a test leakage issue, and WN18RR (Dettmers et al., 2018) has been proposed to resolve the issue. In our experiments, we use WN18RR++ which is the fixed version of WN18RR as described in Section 3.2. Our VTKG datasets include images for triplets and entities, whereas the existing benchmark datasets provide images only for entities.

We use ten different baseline methods: ANAL-OGY (Liu et al., 2017), ComplEx-N3 (Lacroix et al., 2018), RotatE (Sun et al., 2019), PairRE (Chao et al., 2021), RSME (Wang et al., 2021), TransAE (Wang et al., 2019), MKG-former (Chen et al., 2022), OTKGE (Cao et al., 2022), MoSE (Zhao et al., 2022), and IMF (Li et al., 2023). The first four methods are knowledge graph embedding methods only considering the structure of the given knowledge graph, whereas RSME can deal with images of entities. The rest

|  | $|\mathcal{V}|$ | $|\mathcal{R}|$ | $|\mathcal{T}|$ | $|\mathcal{I}|$ | $|\mathcal{D}|$ |
|---|---|---|---|---|---|
| VTKG-I | 181 | 217 | 1,316 | 390,658 | 383 |
| VTKG-C | 43,267 | 2,731 | 111,491 | 461,007 | 45,401 |
| WN18RR++ | 41,105 | 11 | 93,003 | 70,349 | 41,105 |
| FB15K237 | 14,541 | 237 | 310,116 | 145,944 | 14,515 |

Table 2: Statistic of Datasets

five methods are multimodal knowledge graph representation learning methods that consider images and text descriptions of entities. We run MoSE with three different options, denoted by MoSE-AI, MoSE-BI, and MoSE-MI. Details about running the baseline methods are described in Appendix B.

We divide the training, valid, and test sets with a ratio of 8:1:1 for VTKGs; we use the provided split for the existing benchmark datasets. When training a model using triplets' images, we only use the images associated with the triplets in a training set. For knowledge graph completion in the VTKG datasets, if the task is to predict a head or a tail entity in $\langle h, r, t \rangle$, we do not provide any image associated with the triplet $\langle h, r, t \rangle$ to a model at a testing time because its image can directly hint at the missing entity. From training to testing, we never use images of test triplets. We explain the hyperparameters of VISTA in Appendix C. For fair comparisons, we set $d = 256$ on all datasets for all methods except MKGformer; the dimension in MKGformer is fixed to $d = 768$ since it directly uses ViT and BERT.

## 5.2 Knowledge Graph Completion

We evaluate the knowledge graph completion performance using MR ($\downarrow$), MRR ($\uparrow$), and Hit@N ($\uparrow$, N=1,3,10) in Table 3. The best results are boldfaced, and the second-best results are underlined. Overall, VISTA significantly outperforms all baseline methods. While the second-best method varies depending on the dataset and the metric, VISTA consistently shows the best performance. In VTKG-I and VTKG-C, there is a substantial performance gap between VISTA and the best baseline method in all metrics. While VISTA utilizes relations' visual representation vectors and their text descriptions, none of the baseline methods is capable of dealing with them. This difference attributes to the performance gap between VISTA and the baseline methods since the VTKG datasets include visually expressible relations provided by images of triplets and abundant text descriptions for relations. On the other hand, WN18RR++ and

| | VTKG-I | | | | | VTKG-C | | | | |
|---|---|---|---|---|---|---|---|---|---|---|
| | MR | MRR | Hit@1 | Hit@3 | Hit@10 | MR | MRR | Hit@1 | Hit@3 | Hit@10 |
| ANALOGY | 39.5 | 0.3040 | 0.2328 | 0.3015 | 0.4466 | 10392.1 | 0.2963 | 0.2609 | 0.3180 | 0.3532 |
| ComplEx-N3 | 31.8 | 0.3911 | 0.3168 | 0.4046 | 0.5191 | 3668.0 | 0.3944 | 0.3515 | 0.4079 | 0.4815 |
| RotatE | 24.5 | 0.3131 | 0.2099 | 0.3473 | 0.5267 | 4121.2 | 0.3893 | 0.3473 | 0.4062 | 0.4704 |
| PairRE | 20.9 | 0.4104 | 0.3015 | 0.4504 | 0.6145 | 2736.8 | 0.3876 | 0.3431 | 0.4013 | 0.4782 |
| RSME | 32.0 | 0.4027 | 0.3321 | 0.4122 | 0.5573 | 4401.4 | 0.3942 | 0.3513 | 0.4096 | 0.4776 |
| TransAE | 19.5 | 0.2437 | 0.0687 | 0.3092 | 0.6374 | 3063.3 | 0.0751 | 0.0053 | 0.1053 | 0.2072 |
| MKGformer | 29.9 | 0.3884 | 0.3397 | 0.3740 | 0.4885 | 668.5 | 0.4227 | 0.3531 | 0.4487 | 0.5580 |
| OTKGE | 27.5 | 0.4278 | 0.3588 | 0.4466 | 0.5458 | 2606.8 | 0.3939 | 0.3446 | 0.4152 | 0.4881 |
| MoSE-AI | 22.5 | 0.4306 | 0.3473 | 0.4466 | 0.6221 | 854.3 | 0.3929 | 0.3186 | 0.4301 | 0.5210 |
| MoSE-BI | 23.7 | 0.4297 | 0.3473 | 0.4466 | 0.6221 | 615.9 | 0.3933 | 0.2825 | 0.4578 | 0.5864 |
| MoSE-MI | 34.0 | 0.4235 | 0.3397 | 0.4504 | 0.6107 | 527.0 | 0.4056 | 0.2913 | 0.4762 | 0.5977 |
| IMF | 35.6 | 0.4184 | 0.3282 | 0.4656 | 0.5649 | 2951.6 | 0.4116 | 0.3706 | 0.4261 | 0.4935 |
| VISTA | **17.3** | **0.4650** | **0.3626** | **0.5076** | **0.6641** | **220.8** | **0.4675** | **0.3918** | **0.4961** | **0.6157** |
| | WN18RR++ | | | | | FB15K237 | | | | |
| | MR | MRR | Hit@1 | Hit@3 | Hit@10 | MR | MRR | Hit@1 | Hit@3 | Hit@10 |
| ANALOGY | 7453.8 | 0.4128 | 0.3969 | 0.4175 | 0.4438 | 371.9 | 0.2420 | 0.1516 | 0.2677 | 0.4301 |
| ComplEx-N3 | 3545.0 | 0.4745 | 0.4292 | 0.4895 | 0.5675 | 172.7 | 0.3510 | 0.2584 | 0.3847 | 0.5391 |
| RotatE | 5946.1 | 0.4606 | 0.4274 | 0.4754 | 0.5230 | 246.1 | 0.3099 | 0.2183 | 0.3433 | 0.4932 |
| PairRE | 3676.2 | 0.4529 | 0.4127 | 0.4663 | 0.5351 | 184.3 | 0.3326 | 0.2399 | 0.3675 | 0.5193 |
| RSME | 4062.1 | 0.4567 | 0.4175 | 0.4751 | 0.5300 | 174.9 | 0.3445 | 0.2523 | 0.3780 | 0.5286 |
| TransAE | 2308.2 | 0.0900 | 0.0040 | 0.1291 | 0.2511 | 234.6 | 0.2122 | 0.1432 | 0.2267 | 0.3459 |
| MKGformer | 352.3 | 0.5308 | 0.4697 | 0.5557 | 0.6560 | 297.6 | 0.3095 | 0.2278 | 0.3356 | 0.4740 |
| OTKGE | 1993.1 | 0.4327 | 0.3722 | 0.4663 | 0.5407 | 168.1 | 0.3411 | 0.2511 | 0.3739 | 0.5192 |
| MoSE-AI | 303.2 | 0.4857 | 0.4255 | 0.5094 | 0.5996 | 149.1 | 0.3247 | 0.2384 | 0.3532 | 0.4965 |
| MoSE-BI | **108.0** | 0.5026 | 0.4151 | 0.5461 | 0.6670 | 132.3 | 0.3466 | 0.2570 | 0.3755 | 0.5303 |
| MoSE-MI | 205.3 | 0.4969 | 0.4026 | 0.5466 | 0.6681 | 148.9 | 0.3275 | 0.2416 | 0.3568 | 0.4984 |
| IMF | 3774.0 | 0.4749 | 0.4397 | 0.4845 | 0.5469 | 151.8 | 0.3677 | 0.2735 | 0.4040 | 0.5573 |
| VISTA | 177.6 | **0.5526** | **0.4871** | **0.5799** | **0.6755** | **114.2** | **0.3808** | **0.2873** | **0.4158** | **0.5718** |

Table 3: Knowledge Graph Completion Performance. VISTA outperforms all baseline methods in all datasets.

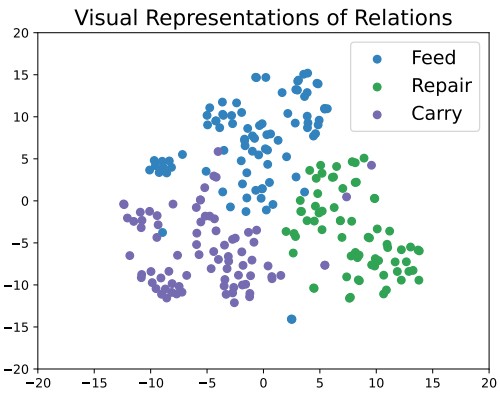

Figure 4: Visual representation vectors of relations, Feed, Repair, and Carry, where the vectors are computed on images of triplets containing the three relations.

FB15K237 do not include images for triplets and descriptions for relations. In WN18RR++, even though VISTA shows the second-best performance in MR, the MRR, Hit@1, Hit@3, and Hit@10 scores of VISTA are better than any other methods. In FB15K237, VISTA outperforms all baseline methods.

## 5.3 Qualitative Analysis

We conduct qualitative analysis using VISTA trained on VTKG-C. In Section 4.3, VISTA learns $\widehat{\mathbf{W}}_{\text{vis}}$ to compute the visual representation vectors of relations from images. Figure 4 shows the visual representation vectors of three relations, Feed, Repair, and Carry, visualized by t-SNE. From a testing set, we select images corresponding to triplets containing the three relations and compute the visual representation vectors using the learned $\widehat{\mathbf{W}}_{\text{vis}}$. We see that the visual representations corresponding to the same relation are well clustered even though VISTA computes them based on unseen images and triplets during training.

We qualitatively compare the representation vectors created by BERT (Devlin et al., 2019), ViT (Dosovitskiy et al., 2021), and VISTA in VTKG-C. Given a query entity (or a relation), we retrieve the top 3 similar entities (or relations) to the query based on the representation vectors returned by BERT, ViT, and VISTA. Table 4 shows the results where the query entities are jar and dark_red, and the query relations are have and buy. We see that BERT returns some abstract con-

| Query | | BERT | ViT | VISTA |
|---|---|---|---|---|
| | 1 | eyelet | lid | bucket |
| jar | 2 | tent | top | barrel |
| | 3 | gravy_boat | coquilla_nut | bowl |
| | 1 | incense | leisure_wear | orange |
| dark_red | 2 | coloring | sportswear | red |
| | 3 | buffer | sweatshirt | crimson |
| | 1 | move | straddle | keep |
| have | 2 | influence | hop_on | hold |
| | 3 | begin | inspect | incorporate |
| | 1 | stipulate | pluck | share |
| buy | 2 | incorporate | jab | price |
| | 3 | use | own | trade |

Table 4: Top three similar entities or relations to the query in VTKG-C based on the representation vectors created by BERT, ViT, and VISTA.

cepts, e.g., `incense` and `influence`, whereas ViT returns visually expressible concepts, e.g., `lid` and `straddle`. VISTA successfully returns the most semantically close entities and relations to the queries by utilizing both texts and images.

As described in Section 4.2, the entity encoding transformer of VISTA learns the representation vector of each entity based on $\mathbf{z}_{[ENT]}$, $\mathbf{h}$, $\mathbf{h}_{vis,1}$, $\cdots$, $\mathbf{h}_{vis,k}$, and $\mathbf{h}_{txt}$. Similarly, the relation encoding transformer learns the representation vector of a relation based on $\mathbf{z}_{[REL]}$, $\mathbf{r}$, $\mathbf{r}_{vis,1}$, $\cdots$, $\mathbf{r}_{vis,\hat{k}}$, and $\mathbf{r}_{txt}$ as described in Section 4.3. To analyze the contribution of each of these terms, we visualize the attention weights using kernel density estimate plots. Figure 5 shows the distributions of the attention weights for entities and relations for (a) the case where the entities and relations do not have images and (b) the case where the entities and relations have images. When we consider the case when images are not given, textual features $\mathbf{r}_{txt}$, play the most crucial role in relations. On the other hand, in entities, the learnable vector $\mathbf{h}$ has relatively high attention weights. When images are given, the visual attributes of relations $\mathbf{r}_{vis,1}$, tend to have high contributions, sometimes even exceeding the contributions of textual descriptions $\mathbf{r}_{txt}$. This verifies that learning visual representations of relations are effective in VISTA, which contributes to the performance of VISTA. For entities, the learnable vector $\mathbf{h}$ still has high importance when images are given, and the visual features $\mathbf{h}_{vis,1}$ tend to have slightly higher importance than textual features $\mathbf{h}_{txt}$.

## 5.4 Ablation Studies

We present ablation studies of VISTA in Table 5. We show the results of VISTA using different com-

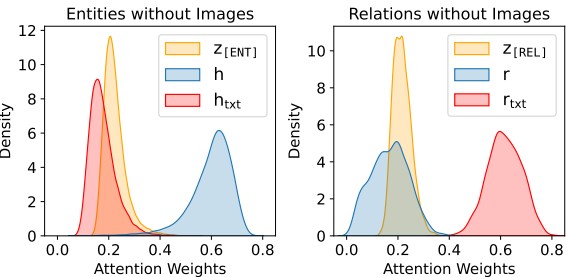

(a) Attention weights in the entity and relation encoding transformers for entities and relations without images

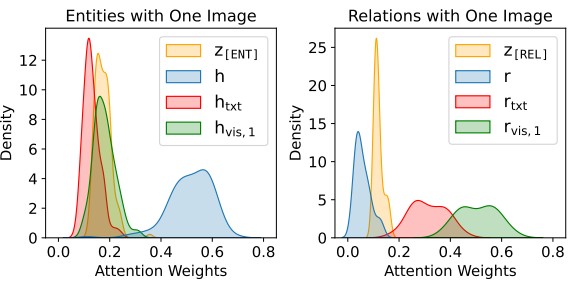

(b) Attention weights in the entity and relation encoding transformers for entities and relations with one image

Figure 5: Distributions of attention weights in the entity and relation encoding transformers.

binations of the modalities (the "Modality" rows). Also, we show the impact of different modalities for entities (the "Entity Modality" rows) and for relations (the "Relation Modality" rows). On both VTKG-I and VTKG-C, we see that adding visual or textual features leads to better performance than the case without those features. More importantly, using all modalities presented in the last row of Table 5 (VISTA) leads to the best performance.

The impact of the visual features is more prominent on VTKG-I than on VTKG-C. To analyze it, we count the number of entities and relations with and without visual features, shown in Table 6. While all entities and relations have their visual features on VTKG-I, many entities and relations do not have visual features on VTKG-C. As a result, visual features are more important than textual features on VTKG-I, while textual features are more important than visual features on VTKG-C.

On the other hand, the performance decreases of VISTA with 'replace $\mathbf{r}_{vis}$ w/ $\mathbf{u}$' indicate that our way of constructing $\mathbf{r}_{vis}$ is more effective than just using the union bounding box of a triplet when creating a visual representation vector of $r$.

Furthermore, we remove or replace some components in VISTA. Specifically, for the "Encoder" part, we consider the cases where (1) we remove $\mathbf{z}_{[ENT]}$ and $\mathbf{z}_{[REL]}$, (2) we remove $\mathbf{h}$ and $\mathbf{r}$, (3) we

| | | VTKG-I | | | | | VTKG-C | | | | |
|---|---|---|---|---|---|---|---|---|---|---|---|
| | | MR | MRR | Hit@1 | Hit@3 | Hit@10 | MR | MRR | Hit@1 | Hit@3 | Hit@10 |
| Modality | w/o $h_{vis}$, $h_{txt}$, $r_{vis}$, $r_{txt}$ | 23.3 | 0.398 | 0.309 | 0.431 | 0.573 | 3163.9 | 0.365 | 0.325 | 0.381 | 0.441 |
| | w/o $h_{vis}$, $r_{vis}$ | 21.0 | 0.442 | 0.347 | 0.454 | 0.645 | 238.7 | 0.456 | 0.383 | 0.481 | 0.601 |
| | w/o $h_{txt}$, $r_{txt}$ | 19.5 | 0.453 | 0.363 | 0.473 | 0.634 | 2641.1 | 0.374 | 0.333 | 0.389 | 0.452 |
| Entity Modality | w/o $h_{vis}$, $h_{txt}$ | 22.2 | 0.397 | 0.294 | 0.439 | 0.592 | 3036.6 | 0.366 | 0.327 | 0.382 | 0.440 |
| | w/o $h_{vis}$ | 20.5 | 0.415 | 0.305 | 0.454 | 0.622 | 229.1 | 0.457 | 0.383 | 0.484 | 0.603 |
| | w/o $h_{txt}$ | 17.2 | 0.459 | 0.359 | 0.496 | 0.664 | 2599.5 | 0.372 | 0.331 | 0.387 | 0.449 |
| Relation Modality | w/o $r_{vis}$, $r_{txt}$ | 18.8 | 0.430 | 0.324 | 0.466 | 0.641 | 235.8 | 0.460 | 0.386 | 0.487 | 0.610 |
| | w/o $r_{vis}$ | 19.0 | 0.455 | 0.359 | 0.492 | 0.641 | 223.7 | 0.466 | 0.391 | 0.493 | 0.613 |
| | w/o $r_{txt}$ | 17.3 | 0.459 | 0.351 | 0.500 | 0.664 | 236.3 | 0.461 | 0.385 | 0.490 | 0.609 |
| | replace $r_{vis}$ w/ $u$ | 18.3 | 0.433 | 0.336 | 0.458 | 0.649 | 318.1 | 0.416 | 0.341 | 0.445 | 0.557 |
| Encoder | w/o $z_{[ENT]}$ and $z_{[REL]}$ | 18.6 | 0.402 | 0.290 | 0.454 | 0.622 | 316.8 | 0.402 | 0.325 | 0.434 | 0.549 |
| | w/o $h$ and $r$ | 19.0 | 0.438 | 0.328 | 0.485 | 0.649 | 250.9 | 0.450 | 0.372 | 0.484 | 0.598 |
| | w/o rel. trans. | 18.4 | 0.417 | 0.302 | 0.473 | 0.622 | 369.3 | 0.389 | 0.312 | 0.420 | 0.538 |
| | w/o ent. trans. | 25.2 | 0.406 | 0.332 | 0.424 | 0.546 | 4368.7 | 0.362 | 0.319 | 0.377 | 0.444 |
| Decoder | w/ DistMult decoder | 21.0 | 0.391 | 0.282 | 0.416 | 0.634 | 420.6 | 0.420 | 0.357 | 0.438 | 0.542 |
| | replace dot prod. w/ lin. | 29.5 | 0.406 | 0.317 | 0.435 | 0.580 | 964.5 | 0.430 | 0.380 | 0.448 | 0.524 |
| VISTA | | 17.3 | 0.465 | 0.363 | 0.508 | 0.664 | 220.8 | 0.467 | 0.392 | 0.496 | 0.616 |

Table 5: Ablation Studies of VISTA. The best performance is achieved when all modalities are considered. The performance of VISTA degrades when we remove any component or replace a component with other alternatives.

| | VTKG-I | VTKG-C |
|---|---|---|
| Entities w/ $h_{vis}$ | 181 (100%) | 7,863 (18.2%) |
| Entities w/o $h_{vis}$ | 0 (0%) | 35,404 (81.8%) |
| Total | 181 | 43,267 |
| Relations w/ $r_{vis}$ | 217 (100%) | 217 (7.9%) |
| Relations w/o $r_{vis}$ | 0 (0%) | 2,514 (92.1%) |
| Total | 217 | 2,731 |

Table 6: Entities and relations with and without visual features on VTKG-I and VTKG-C.

remove the relation encoding transformer and just leave the learnable vector $r$, and (4) we remove the entity encoding transformer and just leave the learnable vector $h$. At the first two rows of the "Encoder" in Table 5, we see that $z_{[ENT]}$, $z_{[REL]}$, $h$, and $r$ are all important inputs of the entity and relation encoding transformers. At the last two rows of the "Encoder" in Table 5, we note that removing either the entity encoding transformer or the relation encoding transformer greatly impacts VISTA's performance. Regarding Hit@1, removing the relation encoding transformer leads to a more critical decrease in performance than removing the entity encoding transformer. On the other hand, for the other metrics (i.e., MR, MRR, Hit@3, and Hit@10), the entity encoding transformer is more critical than the relation encoding transformer.

For the "Decoder" part, we consider the cases where (1) we replace the triplet decoding transformer with DistMult (Yang et al., 2015) and (2) we replace the dot product with a linear layer for prediction. The two rows of the "Decoder" in Table 5 indicate that making changes in the way of predicting a missing entity for knowledge graph completion leads to degrading the performance of VISTA. This validates the effectiveness of our triplet decoding transformer.

## 6 Conclusion & Future Work

We propose VTKGs, where visually expressible triplets are augmented by images, and both entities and relations have textual descriptions. By appropriately utilizing all this rich information, VISTA substantially outperforms 10 different state-of-the-art knowledge graph completion methods in real-world VTKG datasets. Our VTKG datasets and VISTA model can be utilized in diverse applications and scenarios (Sekuboyina et al., 2019; Kwak et al., 2022; Lee et al., 2023), including those requiring visual commonsense knowledge such as VQA (Marino et al., 2021) or scene graph generation (Chang et al., 2023; Zareian et al., 2020) and commonsense reasoning (Lin et al., 2019).

We will extend our work to hyper-relational knowledge graphs (Galkin et al., 2020; Chung et al., 2023) or bi-level knowledge graphs (Chung and Whang, 2023) where more information is added to each triplet using qualifiers, or higher-level relationships are considered to enrich information between triplets. By structuring an image or a description as auxiliary information or qualifiers, VISTA can be easily extended to hyper-relational knowledge graphs with images and texts.

## Limitations

When constructing our VTKG datasets, we aligned the same entities and relations using WordNet synsets (Miller, 1995) as described in Section 3.2; and this process was manually conducted, which required much human effort. While utilizing automated tools would make the process more scalable, we could not find reliable and accurate annotation tools for our alignment task. Developing automated tools that can align triplets from different sources as precisely as human annotators will accelerate the convergence of diversely-sourced knowledge.

Our model, VISTA, comprises three transformers and thus might not be considered as lightweight compared to classical shallow encoder-based knowledge graph embedding methods such as RotatE (Sun et al., 2019) and ComplEx-N3 (Lacroix et al., 2018). However, we note that VISTA is faster than another transformer-based baseline, MKGformer; to process VTKG-C, VISTA took 3 hours, whereas MKGformer took 15 hours. We will explore how we can make our implementations more scalable and make our model lighter.

## Acknowledgements

This research was partly supported by NRF grants funded by MSIT (2022R1A2C4001594 and 2018R1A5A1059921). This work was also supported by an IITP grant funded by MSIT 2022-0-00369 (Development of AI Technology to support Expert Decision-making that can Explain the Reasons/Grounds for Judgment Results based on Expert Knowledge).

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

# Appendix

## A  Details about Constructing and Validating VTKGs

As described in Section 3.2, we examined all triplets in VRD, UnRel, HICO-DET, VisKE, and ConceptNetW to map each entity and relation to a synset in WordNet (Miller, 1995). We developed a data labeling tool shown in Figure 6 that allows a human expert to select the most appropriate synset of each word in a triplet by examining the meaning of each synset and the semantics of the word in context. It took six weeks to examine all the triplets. In the data labeling process, we also verified whether each triplet is valid or not; we removed the invalid or hard-to-interpret triplets from our datasets, e.g., ⟨mechanic, winter, car⟩. During this process, 269 triplets were removed among 19,065 original triplets. When we sampled 2,000 triplets and double-checked the labels, the labeling accuracy was 95.01%.

Additionally, we checked the matching of each triplet and its image. In our datasets, 1,316 triplets are expressed by images, and there are 141,247 images for triplets. The triplet images came from VRD, UnRel, and HICO-DET. To examine whether a triplet is well matched with its image, we randomly sampled 100 images for each triplet. If a triplet has less than 100 images, we examined all the images for the triplet. As a result, we examined 36,919 images, and each image was examined by two different annotators. We consider a triplet is matched with its image only if both annotators agree on it, i.e., if one of the annotators disagrees, we consider the triplet is not matched with its image. When measuring the matching degree in this way, the accuracy was 98.62%.

When constructing our VTKG datasets, we did not use FB15K237 because FB15K237 delivers a different level of information from VRD, UnRel, HICO-DET, VisKE, ConceptNetW, and WN18RR++. For example, while VTKG-C contains generic relationships between concepts, e.g., ⟨person, write, story⟩, most triplets in FB15K237 provide specific details about particular individuals or facts, e.g., ⟨James_Cameron, graduate_from, California_State_University⟩.

We provide some examples of the triplets and their images and descriptions in our VTKG datasets. As explained above, each entity or re-

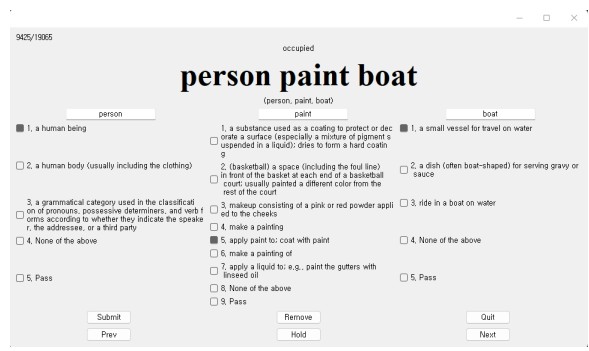

Figure 6: Interface of our data labeling tool for mapping each word to the most appropriate synset and also checking the validity of each triplet.

lation is represented using a synset of WordNet, e.g., 'airplane.n.01' or 'have.v.01', where 'n' indicates a noun, 'v' indicates a verb, and the last two digits indicate the sense number. The descriptions of entities and relations are from WordNet. Figure 7 shows some examples of triplets in our VTKG datasets.

**Merging triplets from different sources**

One of the critical points of creating VTKGs is properly combining triplets from different sources. Since triplets from different sources are usually written using different vocabularies, as discussed in Section 3.2, we had to find a way to consolidate those vocabularies. To resolve this issue, we replace each entity/relation in a triplet with a synset ID defined in WordNet to express triplets using the vocabulary defined in WordNet. An alternative method we considered was to replace each entity/relation with a word defined in the Cambridge Dictionary. However, using the Cambridge Dictionary resulted in creating redundant triplets due to various synonyms, failing to merge semantically identical triplets. Therefore, we decided to use WordNet synsets when we merge triplets.

## B  Experimental Details about Running the Baseline Methods

We run baseline methods using GeForce RTX 2080 Ti or RTX A6000 depending on the dependency of each method. For all methods, we use the default hyperparameters provided in their implementations unless otherwise stated. We choose the best configuration for each method based on the validation results. In what follows, we use the notation of the original papers.

For ANALOGY (Liu et al., 2017), we tried

| ⟨Head Entity, | Relation, | Tail Entity⟩ | Image #1 | Image #2 |
|---|---|---|---|---|
| ⟨**airplane.n.01,** an aircraft that has a fixed wing and is powered by propellers or jets | **have.v.01,** have or possess, either in a concrete or an abstract sense | **wheel.n.01**⟩ a simple machine consisting of a circular frame with spokes (or a solid disc) that can rotate on a shaft or axle (as in vehicles or other machines) | | |
| ⟨**kite.n.03,** plaything consisting of a light frame covered with tissue paper; flown in wind at end of a string | **fly.v.01,** travel through the air; be airborne | **sky.n.01**⟩ the atmosphere and outer space as viewed from the earth | | |
| ⟨**person.n.01,** a human being | **drive.v.01,** operate or control a vehicle | **car.n.01**⟩ a motor vehicle with four wheels; usually propelled by an internal combustion engine | | |

Figure 7: Examples of Triplets in VTKGs

the learning rate $\in \{0.001, 0.005, 0.01, 0.05, 0.1\}$ and $\lambda \in \{0.001, 0.005, 0.01, 0.05, 0.1\}$. We tuned ComplEx-N3 (Lacroix et al., 2018) using the regularization coefficient $\in \{0.0, 0.01, 0.05, 0.1, 0.2\}$. In RotatE (Sun et al., 2019) and PairRE (Chao et al., 2021), we tried the learning rate $\in \{0.00005, 0.0001\}$ and $\gamma \in \{6.0, 9.0, 12.0\}$. RSME (Wang et al., 2021) is tuned with the learning rate $\in \{0.005, 0.01, 0.05\}$ and the regularization coefficient $\in \{0.05, 0.1, 0.15, 0.2\}$. We tuned TransAE (Wang et al., 2019) with $\lambda \in \{0.00005, 0.0001, 0.0005, 0.001, 0.01\}$ and $\gamma \in \{2.0, 2.5, 5.0, 7.5, 10.0\}$. In MKGformer (Chen et al., 2022), we tried the learning rate $\in \{0.00005, 0.0005\}$ in the fine-tuning stage. When we ran MKGformer, we had to decrease the batch size from 96 to 32 due to the memory constraint of our GPUs. For OTKGE (Cao et al., 2022), we tried the batch size $\in \{2000, 5000\}$ and the regularization coefficient $\in \{0.001, 0.005\}$. For MoSE (Zhao et al., 2022) and IMF (Li et al., 2023), we used the default parameters provided in their implementations. The results reported in the original IMF paper are not reproducible.

## C Hyperparameters of VISTA

In VISTA, the learning rate was set to 0.0001 for VTKG-I and VTKG-C, and 0.001 for WN18RR++ and FB15K237. Unless otherwise stated, the hidden dimension of the transformers was fixed to 2,048. We set $L = 2$, $\widehat{L} = 1$, and $k = \widehat{k} = 1$. The dropout rate of the transformers is 0.1, the dropout

rate of the embedding matrices is 0.9, the dropout rate of the textual representation vectors is 0.1, and the step size of the cosine learning rate scheduler is 50. We validated the model performance every 50 epochs.

We tuned VISTA on VTKG-I with $L \in \{2, 3, 4\}$, $\widehat{L} \in \{1, 2, 3\}$, $\widetilde{L} \in \{1, 2\}$, number of attention heads $\in \{2, 4\}$, the dropout rate of the embedding matrices $\in \{0.5, 0.6, 0.7\}$, the dropout rate of the visual representation vectors $\in \{0.3, 0.4\}$, and $k = \widehat{k} \in \{1, 3, 5\}$. We used 768 as the hidden dimension of the transformers, 128 as the batch size, and 0.01 as the dropout rate of the transformers. NVIDIA GeForce RTX 2080 Ti was used to run VISTA on VTKG-I, and it took approximately one minute for a single run with the best hyperparameters.

For VTKG-C, VISTA was tuned with $\widetilde{L} \in \{1, 2\}$, number of attention heads $\in \{2, 4\}$, the dropout rate of the visual representation vectors $\in \{0.3, 0.4\}$, and the dropout rate of the textual representation vectors $\in \{0.0, 0.1\}$. We set the batch size to be 512, $k = \widehat{k} = 3$, and set the dropout rate of the transformers to be 0.01. NVIDIA GeForce RTX 3090 was used to run VISTA on VTKG-C, and it took approximately three hours for a single run with the best hyperparameters.

We tuned VISTA on WN18RR++ with number of attention heads $\in \{8, 16\}$, the dropout rate of the visual representation vectors $\in \{0.2, 0.3\}$, the dropout rate of the textual representation vectors $\in \{0.0, 0.1\}$, and the step size of the cosine learning rate scheduler $\in \{50, 100\}$. We set $\widetilde{L} = 1$ and

the batch size is 1024. We used NVIDIA GeForce RTX 3090 and NVIDIA RTX A6000 to run VISTA on WN18RR++, and it took approximately 6 hours for a single run with the best hyperparameters.

For FB15K237, VISTA was tuned with $L \in \{1, 2\}$, number of attention heads $\in \{32, 64\}$, the dropout rate of the embedding matrices $\in \{0.8, 0.9\}$, the step size of the cosine learning rate scheduler $\in \{50, 100\}$, and the batch size $\in \{512, 1024\}$. We set the dropout rate of the visual representation vectors and the textual representation vectors to 0.3 and 0.0, respectively, and $\widetilde{L} = 1$. We used NVIDIA GeForce RTX 3090 and NVIDIA RTX A6000 to run VISTA on FB15K237, and it took approximately 15 hours for a single run with the best hyperparameters.