# OpenReview forum: "VISTA: Visual-Textual Knowledge Graph Representation Learning"
_EMNLP/2023/Conference — EMNLP 2023 Findings_

### Official Review · Reviewer_V1gZ · 2023-08-03

**Soundness:** 3

**Excitement:**

3: Ambivalent: It has merits (e.g., it reports state-of-the-art results, the idea is nice), but there are key weaknesses (e.g., it describes incremental work), and it can significantly benefit from another round of revision. However, I won't object to accepting it if my co-reviewers champion it.

**Paper Topic And Main Contributions:**

 This paper constructs new benchmark datasets for multimodal knowledge graph, which contains visually expressible commonsense knowledge, the images of entities and triples and detailed descriptions of entities and relations. And a visual-textual (VISTA) method is proposed to for knowledge graph representation learning.

**Reasons To Accept:**

 1. New benchmark datasets for multimodal knowledge graph, which is useful for studies on multimodal knowledge graph.
 2. A visual-textual (VISTA) method is proposed and extensive experiments are conducted on the new datasets.

**Reasons To Reject:**

The experiments about impact of different modal information for different baseline models are required to better understand the benefits of multimodal KG.

**Reproducibility:**

2: Would be hard pressed to reproduce the results. The contribution depends on data that are simply not available outside the author's institution or consortium; not enough details are provided.

**Reviewer Confidence:**

3: Pretty sure, but there's a chance I missed something. Although I have a good feel for this area in general, I did not carefully check the paper's details, e.g., the math, experimental design, or novelty.

---

> ### Author Rebuttal · Authors · 2023-08-28
>
> We appreciate your time and effort in reading our paper. In what follows, we provide our answers to your concerns. Please also consider that we prepared our answers to **seven different reviewers** during the rebuttal period. By incorporating the comments from seven reviewers, including yours, we got a chance to improve our paper. We hope the improvements can be considered in the final assessment of our paper.
>
> $\rule{18cm}{0.05cm}$
>
> > The experiments about impact of different modal information for different baseline models are required to better understand the benefits of multimodal KG.
>
> Among the baseline methods, OTKGE[1], MoSE[2], and IMF[3] presented the impact of different modal information in their original papers, showing that utilizing different modalities is beneficial to improving the performance of knowledge graph completion. To analyze the impact of each modality in VISTA, we added two tables below. In Table A1, we show the results of VISTA using different combinations of the modalities. In VISTA, we consider $\bf{h}$ in the entity encoding transformer and $\bf{r}$ in the relation encoding transformer as the structural representations of an entity and a relation, respectively (Figure 3). On both VTKG-I and VTKG-C, we see that adding visual or textual features leads to better performance than the case without those features. We also note that the impact of the visual features is more prominent on VTKG-I than on VTKG-C. To understand this point more thoroughly, we also analyze the statistics about categorizing triplets based on whether their entities and relations have visual features, shown in Table A2. While all entities and relations have their visual features on VTKG-I, many entities and relations do not have visual features on VTKG-C. This explains why visual features are more critical on VTKG-I. More importantly, **Table A1 shows using all the modalities (structural+textual+visual) leads to the best performance of VISTA**.
>
> ### **Table A1: Performance of VISTA using different combinations of modalities**
>
> $\begin{array}{|c|ccccc|ccccc|}
> \hline
> &&&\hspace{21mu}\textclap{VTKG-I}&&&&&\hspace{6mu}\textclap{VTKG-C}&& \\\\
> &\text{MR} & \text{MRR} & \text{Hit@1} & \text{Hit@3} & \text{Hit@10} & \text{MR} & \text{MRR} & \text{Hit@1} &  \text{Hit@3} & \text{Hit@10} \\\\
> \hline
> \text{structural} & 23.3 & 0.3979 & 0.3092 &0.4313 & 0.5725 & 3163.9 &0.3651 & 0.3252 & 0.3806 & 0.4406 \\\\
> \text{structural + textual} & 21.0& 0.4423& 0.3473& 0.4542& 0.6450 & 238.7& 0.4556& 0.3827& 0.4811& 0.6014 \\\\
> \text{structural + visual} & 19.5& 0.4525& 0.3626& 0.4733& 0.6336  & 2641.1& 0.3743& 0.3331& 0.3893& 0.4523 \\\\
> \hline
> \text{structural + textual + visual} & \bf{17.3} & \bf{0.4650} & \bf{0.3626} & \bf{0.5076} & \bf{0.6641} & \bf{220.8} & \bf{0.4675} & \bf{0.3918} & \bf{0.4961} & \bf{0.6157} \\\\
> \hline
> \end{array}$
>
> ### **Table A2: Categorizing triplets based on whether their entities and relations have visual features**
>
> $\begin{array}{rcc}
> \hline
>  & \text{VTKG-I} & \text{VTKG-C} \\\\
> \hline
> \text{two entities w/ visual features \\& relation w/ visual features} & 1,316\ (100\\%) & 2,569\ (2.3\\%) \\\\
> \text{one entity w/ visual features \\& relation w/ visual features} & 0\ (0\\%) & 1,743\ (1.6\\%) \\\\
> \text{two entities w/o visual features \\& relation w/ visual features} & 0\ (0\\%) & 522\ (0.5\\%) \\\\
> \hline
> \text{two entities w/ visual features \\& relation w/o visual features} & 0\ (0\\%) & 10,166\ (9.1\\%) \\\\
> \text{one entity w/ visual features \\& relation w/o visual features} & 0\ (0\\%) & 21,106\ (18.9\\%) \\\\
> \text{two entities w/o visual features \\& relation w/o visual features} & 0\ (0\\%) & 75,385\ (67.6\\%) \\\\
> \hline
> \text{Total} & 1,316 & 111,491 \\\\
> \hline
> \end{array}$
>
> **References:**\
> [1] Z. Cao et al., OTKGE: Multi-modal Knowledge Graph Embeddings via Optimal Transport, NeurIPS 2022\
> [2] Y. Zhao et al., MoSE: Modality Split and Ensemble for Multimodal Knowledge Graph Completion, EMNLP 2022\
> [3] X. Li et al., IMF: Interactive Multimodal Fusion Model for Link Prediction, TheWebConf 2023

---

### Official Review · Reviewer_pVWp · 2023-08-03

**Soundness:** 2

**Excitement:**

3: Ambivalent: It has merits (e.g., it reports state-of-the-art results, the idea is nice), but there are key weaknesses (e.g., it describes incremental work), and it can significantly benefit from another round of revision. However, I won't object to accepting it if my co-reviewers champion it.

**Paper Topic And Main Contributions:**

The paper proposes visual-textual knowledge graphs (VTKG) in which the entity, relation, and triple all associate textual descriptions and images, while the existing multimodal knowledge graph (MKGs) methods do not consider the triple images. The paper proposes new datasets for VTKG named VTKG-I and VTKG-C. Moreover, the paper proposes a transformer-based VTKG representation learning method VISTA. The experiments on both VTKG datasets and MKG datasets demonstrate the effectiveness of the method.

**Questions For The Authors:**

1) See Reasons to Reject 1), 2), and 3).

2) How is the ablation study "w/o $z_{ENT}$ and $z_{REL}$" conducted? After removing them, what is the output of the encoder? What is the input of the decoder and Is it only $z_{[MASK]}$? Where is the $z_{[MASK]}$ from? What is the input and the model when obtaining $z_{[MASK]}$?

**Reasons To Accept:**

1) The association of triple images in MKGs is novel.

2) The dataset construction is solid.

3) The experiments on VTKG datasets and MKG datasets outperform the state-of-the-art baselines.

**Reasons To Reject:**

1) The VTKG experiment setting with triple images may lack practical application. The paper utilizes triple images during training, which are not available during testing. If the triplet image is available in testing, the two entities that appeared in one image directly reveal the ground truth and do not need further inference. If they are not available, I could not think of one application scenario. Moreover, the KGs organize the entities and relations separately and the triple image may be contrary to it.

2) How VISTA work in the test phase is ambiguous. In the test phase, how is the relation embedding and $r_{vis}$ calculated? Is the input of the relation encoder only $r_{txt}$? If the $r_{vis}$ is calculated by concatenating head and tail images, there may be information leakage. I think the decline of "replace $r_{vis}$ w/ $u$" may confirm the problem.

3) How VISTA work in MKGs without triple images is not clear. How are $r_{vis}$ and $u$ calculated in the FB15K-237 and WN18RR++ datasets?

**Reproducibility:**

3: Could reproduce the results with some difficulty. The settings of parameters are underspecified or subjectively determined; the training/evaluation data are not widely available.

**Reviewer Confidence:**

4: Quite sure. I tried to check the important points carefully. It's unlikely, though conceivable, that I missed something that should affect my ratings.

**Typos Grammar Style And Presentation Improvements:**

Writing needs improvement. Some details of the method and the experiment setting are not clear. The notations in Section 4 are messy. It would be easier to read to simplify some notations.

---

> ### Author Rebuttal · Authors · 2023-08-28
>
> We appreciate your time and effort in reading our paper. In what follows, we provide our answers to your concerns. Please also consider that we prepared our answers to **seven different reviewers** during the rebuttal period. By incorporating the comments from seven reviewers, including yours, we got a chance to improve our paper. We hope the improvements can be considered in the final assessment of our paper. We believe most of your concerns may be resolved by the clarifications below. In particular, please read our first two responses together since they answer closely related questions.
>
> $\rule{18cm}{0.05cm}$
>
> > The VTKG experiment setting with triple images may lack practical application. The paper utilizes triple images during training, which are not available during testing. If the triplet image is available in testing, the two entities that appeared in one image directly reveal the ground truth and do not need further inference. If they are not available, I could not think of one application scenario. Moreover, the KGs organize the entities and relations separately and the triple image may be contrary to it.
>
> Let us clarify how we extract and utilize the visual features of relations. For an image representing $\langle s,r,o \rangle$, we define a visual feature of relation $r$ as $\bf{r}\_\text{vis}=[\bf{s}\;\bf{o}\;\bf{u}]$ (Lines 270-281). For each relation, we select the top $\hat{k}$ images in the training set based on the pairwise dot product similarities of the $\bf{r}\_\text{vis}$ values (Lines 283-285). When we select the $\hat{k}$ visual features of a relation $r$, i.e., $\bf{r}\_{\text{vis},1}, \cdots, \bf{r}\_{\text{vis},\hat{k}}$, we consider all training triplet images containing the relation $r$ regardless of their head or tail entities. Since a learnable projection matrix $\widehat{\bf W}\_\text{vis}$ is applied to each $\bf{r}\_\text{vis}$ vector to construct an initial visual representation vector of $r$, the $\bf{r}\_\text{vis}$ vectors do not need to share $s$ or $o$. Once the initial visual representation vectors of relations (e.g., $\widehat{\bf W}\_\text{vis} \bf{r}\_{\text{vis},1}$) are fed into the relation encoding transformer (Figure 3), the final representations of relations are the ones that reflect the visual features extracted from training triplet images. **Note that we never use images of test triplets, as we already explained in Lines 412-423.** Using the final representations of entities and relations, we predict a missing entity in a triplet at testing time. While VISTA is different from existing methods in that it can incorporate the visual and textual features of relations when learning the representations of relations during training, our prediction task itself is the same as a general multimodal knowledge graph completion problem and does not need a separate application scenario.
>
> $\rule{18cm}{0.05cm}$
>
> > How VISTA work in the test phase is ambiguous. In the test phase, how is the relation embedding and $r_{vis}$ calculated? Is the input of the relation encoder only $r_{txt}$? If the $r_{vis}$ is calculated by concatenating head and tail images, there may be information leakage. I think the decline of "replace $r_{vis}$ w/ $u$" may confirm the problem.
>
> The representation vector of a relation is computed by using the textual feature of $r$ (Lines 286-295) and at most $\hat{k}$ different $\bf{r}\_\text{vis}$ vectors extracted from the training set. **We do not use the images from the test set, so there is no information leakage.** Note that there is no overlap between the training and testing triplets. Regarding how we compute and utilize the $\bf{r}\_\text{vis}$ vectors, please refer to our response to the first comment described above. We extract the visual features of relations from images of training triplets, and the final representation vectors of the relations reflect the training visual features. We will further clarify this point in a revised paper. The experiment of “replace $\bf{r}\_\text{vis}$ w/ $\bf{u}$” is the ablation study about our way of constructing the visual features of relations and has nothing to do with information leakage.
>
> $\rule{18cm}{0.05cm}$
>
> > How VISTA work in MKGs without triple images is not clear. How are $r_{vis}$ and $u$ calculated in the FB15K-237 and WN18RR++ datasets?
>
> Since FB15K237 and WN18RR++ do not have images for triplets (Lines 393-394), ${\bf r}\_\text{vis}$ and $\bf{u}$ are not calculated. For relations without visual features, the representations are learned based on $\bf{z}\_\text{[REL]}$, $\bf{r}$, and $\bf{r}\_\text{txt}$ in the relation encoding transformer. For entities without visual features, the representations are learned based on $\bf{z\_\text{[ENT]}}$, $\bf{h}$, and $\bf{h}\_\text{txt}$ in the entity encoding transformer.
>
> $\rule{18cm}{0.05cm}$
>
> > How is the ablation study "w/o $z_{ENT}$ and $z_{REL}$" conducted? After removing them, what is the output of the encoder? What is the input of the decoder and Is it only $z_{[MASK]}$? Where is the $z_{[MASK]}$ from? What is the input and the model when obtaining $z_{[MASK]}$?
>
> In the ablation study "w/o $\bf{z}\_\text{ENT}$ and $\bf{z}\_\text{REL}$", we use $\bf{h}^{(L)}$ and $\bf{r}^{(\hat{L})}$ for the outputs of the encoding transformers. To solve a problem $\langle h,r,? \rangle$, we use the representation of $h$ returned from the entity encoding transformer, the representation of $r$ returned from the relation encoding transformer, and $\bf{z}\_\text{[MASK]}$ (Lines 353-370); these are the inputs of the triplet decoding transformer. We employ a standard masking scheme in a transformer, and $\bf{z}\_\text{[MASK]}$ is a learned vector during training representing a missing entity in a triplet (Line 355).
>
> $\rule{18cm}{0.05cm}$
>
> > Writing needs improvement. Some details of the method and the experiment setting are not clear. The notations in Section 4 are messy. It would be easier to read to simplify some notations.
>
> We will refine and revise the writing by incorporating all the comments from seven reviewers. We will elaborate more on how we construct $\bf{r}\_\text{vis}$ and the details about the ablation studies and the problem setting. Also, we will add more details about how VISTA works when images are not available. For notations, even though the notations in Section 4 may seem messy, we believe every notation is needed to describe our model precisely. If you could let us know a specific notation that is redundant, we will simplify it.

---

### Official Review · Reviewer_iJiR · 2023-08-11

**Soundness:** 5

**Excitement:**

5: Transformative: This paper is likely to change its subfield or computational linguistics broadly. It should be considered for a best paper award. This paper changes the current understanding of some phenomenon, shows a widely held practice to be erroneous in someway, enables a promising direction of research for a (broad or narrow) topic, or creates an exciting new technique.

**Missing References:**

The worked reminded me a bit to this AKBC paper:

Daniel Oñoro-Rubio, Mathias Niepert, Alberto García-Durán, Roberto Gonzalez-Sanchez, Roberto Javier López-Sastre:
Answering Visual-Relational Queries in Web-Extracted Knowledge Graphs. AKBC 2019

For the Multimodal Knowledge Graph Completion section:
Ye Liu, Hui Li, Alberto García-Durán, Mathias Niepert, Daniel Oñoro-Rubio, David S. Rosenblum:
MMKG: Multi-modal Knowledge Graphs. ESWC 2019: 459-474

Another visual KG application in medical domain:
Anjany Sekuboyina, Daniel Oñoro-Rubio, Jens Kleesiek, Brandon Malone:
A Relational-Learning Perspective To Multi-Label Chest X-Ray Classification. ISBI 2021: 1618-1622



**Paper Topic And Main Contributions:**

The paper VISTA and VTKG. VTKG is a new dataset that builds a KG in which the entities could be represented by text and/or images. VISTA is a new multi-modal model that is designed to solve the problem proposed. They proposed a complete experimental setup. The results shows consistency, and they seems to achieve the state of the art.

**Questions For The Authors:**

Q1: while reading section 4, the operator || appears constantly. I understand it as an element list separator. Am I wrong? Maybe Figure 3 could show the parts that corresponds to some of the equation (e.g. X^(0)). An idea could be to put a color shade that highlight the elements that corresponds to X^(0), X^L,...
Q2: I'm missing a table that shows the final statistics of VTKG. I can see table 2 does the same for the DB that were use as sources, but does it means that VTKG is just the straight combination of them?



**Reasons To Accept:**

The paper could provide to the community a new cross-domain benchmark. The paper is well written, the model and datasets seems technically good. The experimental setup is reasonable, and the results are good.

**Reasons To Reject:**

If the code of the model and the dataset is not release, it could be difficult to replicate the results. However, this alone wouldn't be a strong reason to reject the paper.

**Reproducibility:**

5: Could easily reproduce the results.

**Reviewer Confidence:**

5: Positive that my evaluation is correct. I read the paper very carefully and I am very familiar with related work.

---

> ### Author Rebuttal · Authors · 2023-08-28
>
> We appreciate your time and effort in reading our paper. In what follows, we provide our answers to your concerns. Please also consider that we prepared our answers to **seven different reviewers** during the rebuttal period. By incorporating the comments from seven reviewers, including yours, we got a chance to improve our paper. We hope the improvements can be considered in the final assessment of our paper.
>
> $\rule{15cm}{0.05cm}$
>
> > If the code of the model and the dataset is not release, it could be difficult to replicate the results. However, this alone wouldn't be a strong reason to reject the paper.
>
> We will release all our datasets and codes if this paper is accepted. In what follows, we provide some examples of the triplets and their images and descriptions in our VTKG datasets. As explained in the paper (Lines 225-246), each entity or relation is represented using a synset of WordNet, e.g., ‘airplane.n.01’ or ‘have.v.01’, where ‘n’ indicates a noun, ‘v’ indicates a verb, and the last two digits indicate the sense number. The descriptions of entities and relations are from WordNet. Since we cannot use an external link here, we provide the image IDs of the source dataset for the triplet images.
>
> ____
> ### **(airplane.n.01, have.v.01, wheel.n.01)**
>
> #### **[Descriptions]**
>
> **airplane.n.01**: an aircraft that has a fixed wing and is powered by propellers or jets\
> **have.v.01**: have or possess, either in a concrete or an abstract sense\
> **wheel.n.01**: a simple machine consisting of a circular frame with spokes (or a solid disc) that can rotate on a shaft or axle (as in vehicles or other machines)
>
> #### **[Images]**
>
> 5406472568_78d687217f_b.jpg, 6994124346_d4813d9602_b.jpg, 8059265370_6843fc7ba4_b.jpg from **VRD**
>
> ____
>
> ### **(kite.n.03, fly.v.01, sky.n.01)**
>
> #### **[Descriptions]**
>
> **kite.n.03**: plaything consisting of a light frame covered with tissue paper; flown in wind at end of a string\
> **fly.v.01**: travel through the air; be airborne\
> **sky.n.01**: the atmosphere and outer space as viewed from the earth
>
> #### **[Images]**
>
> 4433556373_ddfabedff5_o.jpg, 8642248228_d2187daf86_b.jpg, 9759509291_143489b32a_b.jpg from **VRD**
>
> ____
>
> ### **(person.n.01, drive.v.01, car.n.01)**
>
> #### **[Descriptions]**
>
> **person.n.01**: a human being\
> **drive.v.01**: operate or control a vehicle\
> **car.n.01**: a motor vehicle with four wheels; usually propelled by an internal combustion engine
>
> #### **[Images]**
>
> HICO_train2015_00002944.jpg, HICO_test2015_00005622.jpg from **HICO-DET**\
> 3141015389_d1dd4fa2ee_b.jpg from **VRD**
>
> ____
>
> $\rule{15cm}{0.05cm}$
>
> > Q1: while reading section 4, the operator || appears constantly. I understand it as an element list separator. Am I wrong? Maybe Figure 3 could show the parts that corresponds to some of the equation (e.g. X^(0)). An idea could be to put a color shade that highlight the elements that corresponds to X^(0), X^L,...
>
> You understood it correctly. We will revise Figure 3 as you suggested. Thank you for the suggestion!
>
> $\rule{15cm}{0.05cm}$
>
> > Q2: I'm missing a table that shows the final statistics of VTKG. I can see table 2 does the same for the DB that were use as sources, but does it means that VTKG is just the straight combination of them?
>
> The first two rows of Table 2 show the final statistics of VTKGs, and Table 1 shows the statistics of the source datasets of VTKGs. Our VTKG datasets are not straight combinations of the source datasets; how we merged the datasets is described in Lines 225-248.
>
> $\rule{15cm}{0.05cm}$
>
> > The worked reminded me a bit to this AKBC paper:\
> Daniel Oñoro-Rubio, Mathias Niepert, Alberto García-Durán, Roberto Gonzalez-Sanchez, Roberto Javier López-Sastre: Answering Visual-Relational Queries in Web-Extracted Knowledge Graphs. AKBC 2019\
> For the Multimodal Knowledge Graph Completion section: Ye Liu, Hui Li, Alberto García-Durán, Mathias Niepert, Daniel Oñoro-Rubio, David S. Rosenblum: MMKG: Multi-modal Knowledge Graphs. ESWC 2019: 459-474\
> Another visual KG application in medical domain: Anjany Sekuboyina, Daniel Oñoro-Rubio, Jens Kleesiek, Brandon Malone: A Relational-Learning Perspective To Multi-Label Chest X-Ray Classification. ISBI 2021: 1618-1622
>
> Thank you for the suggestions! We will mention the first two papers in Section 2, explaining how our work differs from theirs: while their works assume that only entities have images, our VTKGs assume that triplets can also be represented using images, and the relations have visual/textual features. We will mention the last paper in Section 6, describing possible applications of our work.

---

### Official Review · Reviewer_6NUN · 2023-08-11

**Soundness:** 3

**Excitement:**

3: Ambivalent: It has merits (e.g., it reports state-of-the-art results, the idea is nice), but there are key weaknesses (e.g., it describes incremental work), and it can significantly benefit from another round of revision. However, I won't object to accepting it if my co-reviewers champion it.

**Paper Topic And Main Contributions:**

This paper proposes a method VISTA for visual-textual knowledge graphs and conducts new VTKGs benchmark datasets. Extensive experiments validate the effectiveness of VISTA.

**Questions For The Authors:**

● At line 290, does the [CLS] is the last token?
● Why do you put the implementation details in section 4, not section 5, about lines 371-375?
● What do the up and down arrows mean at lines 430-431?

**Reasons To Accept:**

● This paper proposes a VTKG representation learning method.
● The SOTA performance on various datasets.
● The description of the experiment is detailed.
● The figures of the paper provide effective explanations for the idea.

**Reasons To Reject:**

● There are no demo datasets provided.
● Some parts of the paper writing are not rigorous and incoherent.

**Reproducibility:**

2: Would be hard pressed to reproduce the results. The contribution depends on data that are simply not available outside the author's institution or consortium; not enough details are provided.

**Reviewer Confidence:**

3: Pretty sure, but there's a chance I missed something. Although I have a good feel for this area in general, I did not carefully check the paper's details, e.g., the math, experimental design, or novelty.

**Typos Grammar Style And Presentation Improvements:**

● Incoherent descriptions appear at lines 251-257.
● improper title for Section 5.
● The caption of Figure 4 is ambiguous and hard to understand.

---

> ### Author Rebuttal · Authors · 2023-08-28
>
> We appreciate your time and effort in reading our paper. In what follows, we provide our answers to your concerns. Please also consider that we prepared our answers to **seven different reviewers** during the rebuttal period. By incorporating the comments from seven reviewers, including yours, we got a chance to improve our paper. We hope the improvements can be considered in the final assessment of our paper.
>
> $\rule{18cm}{0.05cm}$
>
> > There are no demo datasets provided.
>
> We provide some examples of the triplets and their images and descriptions in our VTKG datasets. As explained in the paper (Lines 225-246), each entity or relation is represented using a synset of WordNet, e.g., ‘airplane.n.01’ or ‘have.v.01’, where ‘n’ indicates a noun, ‘v’ indicates a verb, and the last two digits indicate the sense number. The descriptions of entities and relations are from WordNet. Since we cannot use an external link here, we provide the image IDs of the source dataset for the triplet images. If our paper is accepted, we will release the full datasets and codes.
>
> ----------------
>
> ### **(airplane.n.01, have.v.01, wheel.n.01)**
>
> #### **[Descriptions]**
>
> **airplane.n.01**: an aircraft that has a fixed wing and is powered by propellers or jets\
> **have.v.01**: have or possess, either in a concrete or an abstract sense\
> **wheel.n.01**: a simple machine consisting of a circular frame with spokes (or a solid disc) that can rotate on a shaft or axle (as in vehicles or other machines)
>
> #### **[Images]**
>
> 5406472568_78d687217f_b.jpg, 6994124346_d4813d9602_b.jpg, 8059265370_6843fc7ba4_b.jpg from **VRD**
>
> ____
>
> ### **(kite.n.03, fly.v.01, sky.n.01)**
>
> #### **[Descriptions]**
>
> **kite.n.03**: plaything consisting of a light frame covered with tissue paper; flown in wind at end of a string\
> **fly.v.01**: travel through the air; be airborne\
> **sky.n.01**: the atmosphere and outer space as viewed from the earth
>
> #### **[Images]**
>
> 4433556373_ddfabedff5_o.jpg, 8642248228_d2187daf86_b.jpg, 9759509291_143489b32a_b.jpg from **VRD**
>
> ____
>
> ### **(person.n.01, drive.v.01, car.n.01)**
>
> #### **[Descriptions]**
>
> **person.n.01**: a human being\
> **drive.v.01**: operate or control a vehicle\
> **car.n.01**: a motor vehicle with four wheels; usually propelled by an internal combustion engine
>
> #### **[Images]**
>
> HICO_train2015_00002944.jpg, HICO_test2015_00005622.jpg from **HICO-DET**\
> 3141015389_d1dd4fa2ee_b.jpg from **VRD**
>
> ____
>
> $\rule{18cm}{0.05cm}$
>
> > Some parts of the paper writing are not rigorous and incoherent.
>
> While any specific suggestions would be appreciated, we can confirm that we will refine and revise our paper by incorporating the comments suggested by all reviewers.
>
> $\rule{18cm}{0.05cm}$
>
> > At line 290, does the [CLS] is the last token?
>
> [CLS] is the first token of a sequence. In Line 290, by saying “the last hidden state”, we meant “the hidden state at the last layer”. We will paraphrase the expression.
>
> $\rule{18cm}{0.05cm}$
>
> > Why do you put the implementation details in section 4, not section 5, about lines 371-375?
>
> As you suggested, we will move the implementation details (Lines 371-375) to Section 5.
>
> $\rule{18cm}{0.05cm}$
>
> > What do the up and down arrows mean at lines 430-431?
>
> In Lines 430-431, the down arrow next to MR indicates that a lower MR implies better performance, whereas the up arrows next to MRR and Hit@N indicate that higher MRR and Hit@N imply better performance.
>
> $\rule{18cm}{0.05cm}$
>
> > Incoherent descriptions appear at lines 251-257.
>
> Would you mind letting us know what kind of incoherence you observed in Lines 251-257?
>
> $\rule{18cm}{0.05cm}$
>
> > improper title for Section 5.
>
> We will rename the title of Section 5 as “Experiments”. Other suggestions would also be welcomed if available.
>
> $\rule{18cm}{0.05cm}$
>
> > The caption of Figure 4 is ambiguous and hard to understand.
>
> To show Figure 4, we consider three relations, $\texttt{Feed}$, $\texttt{Repair}$, and $\texttt{Carry}$, and select images representing triplets having those three relations in the test set. Among these selected images, assume that the $j$-th image contains $\texttt{Feed}$. Then, we compute $\widehat{\bf{W}}\_\text{vis} \bf{r}\_{\text{vis},j}$ which is the visual representation vector of $\texttt{Feed}$ for the $j$-th image. In this way, we compute the visual representation vectors of the three relations on the selected images and visualize them in Figure 4 (Lines 458-469). Note that $\widehat{\bf{W}}\_\text{vis}$ is the projection matrix learned during training that converts visual features to visual representation vectors of relations (Lines 320-334). We used the images of the test triplets (those having the above three relations) only for visualization in Figure 4; we never used images of test triplets from training to testing of VISTA, as described in Lines 417-423. We will add more details to the caption of Figure 4.

---

### Official Review · Reviewer_bFL1 · 2023-08-11

**Soundness:** 3

**Excitement:**

3: Ambivalent: It has merits (e.g., it reports state-of-the-art results, the idea is nice), but there are key weaknesses (e.g., it describes incremental work), and it can significantly benefit from another round of revision. However, I won't object to accepting it if my co-reviewers champion it.

**Paper Topic And Main Contributions:**

The authors of this paper present a novel approach aimed at enhancing the multi-modal knowledge graph by incorporating image information into each triplet. To assemble such visual-textual knowledge graph (VTKG), they leverage both computer vision (CV) datasets and commonsense knowledge graphs (CSKG). To be specific, they extract images and corresponding visual phrases (triplets) from the datasets in different computer vision tasks. And then, they rewrite and complement triplets of two large commonsense knowledge graphs, ConceptNet and WN18RR. Finally, they manually merge the triplet from both CV datasets and CSKG according to a group of pre-defined sunsets.

They also design a method of learning representations in visual-textual knowledge graph (VISTA). It first separately encodes visual (image) and textual descriptions with ViT and BERT respectively. With the visual and textual features extracted, they utilize transformer-based encoders to merge multi-source information (visual, textual and triplet) and then learn to predict the missing entities with a triplet decoder. They conduct knowledge graph completion (KGC) task to evaluate their VISTA with VTKG and other knowledge graphs. Experiment results demonstrate their VISTA outperforms the compared models in both multi-modal and traditional knowledge graph. They also conduct qualitative analysis to further validate and explain the effectiveness of VISTA.

**Reasons To Accept:**

1. The idea of expressing the triplets with their images sounds novel and useful. This approach empowers the multi-modal knowledge graph to encompass richer and more comprehensive information, thereby potentially benefiting downstream tasks like visual question answering.
2. Their VISTA achieves promising results in both multi-modal and traditional knowledge graph completion tasks. The subsequent qualitative analysis makes their method more direct and explainable.

**Reasons To Reject:**

1. The presentation of this paper can be improved. Section 3.2 contains a long description of how they create VTKGs, without any sub-heading or text in bold, making others hard to capture the key points in their dataset constructing process. The discussion of VISTA’s each component in section 5.4 is too brief, considering VISTA is quite complex with many different components. Conclusions and analysis like which component play a more critical role in VISTA should be concluded.

2. The quality of VTKG is unclear. In this paper, rather than building a new multi-modal knowledge graph from scratch, the authors design a pipeline to extract and merge triplets from CV and CSKG datasets. This kind of method may cause error accumulation in each step of the pipeline and thus reducing the overall quality of the final knowledge graph. Additionally, both evaluation of VTKG (E.g. matching degree of triplet and its image) and validation of each step in their pipeline are missing in this paper. All these things above make me worried about the quality of their constructed VTKG.

3. They do not clearly validate the effectiveness of introducing image information of triplet into the multimodal knowledge graph. One of the most important contributions they make in this paper is proposing to express triplet in knowledge graph with its image. However, they fail to quantitatively validate the effectiveness of doing so in the experiment section. I suggest adding a new ablation study to train each model w/ and w/o visual information of triplet. They could also consider training models in VTKG and another existing multi-modal knowledge graph respectively, and then testing in a new knowledge graph dataset (out-of-domain setting).

**Reproducibility:**

3: Could reproduce the results with some difficulty. The settings of parameters are underspecified or subjectively determined; the training/evaluation data are not widely available.

**Reviewer Confidence:**

3: Pretty sure, but there's a chance I missed something. Although I have a good feel for this area in general, I did not carefully check the paper's details, e.g., the math, experimental design, or novelty.

---

> ### Author Rebuttal · Authors · 2023-08-28
>
> We appreciate your time and effort in reading our paper. In what follows, we provide our answers to your concerns. Please also consider that we prepared our answers to **seven different reviewers** during the rebuttal period. By incorporating the comments from seven reviewers, including yours, we got a chance to improve our paper. We hope the improvements can be considered in the final assessment of our paper. For better readability, we split and reordered the comments without missing any.
>
> $\rule{18cm}{0.05cm}$
>
> > The presentation of this paper can be improved. Section 3.2 contains a long description of how they create VTKGs, without any sub-heading or text in bold, making others hard to capture the key points in their dataset constructing process.
>
> We will add sub-headings and text in bold to let readers better catch the overall procedure of our dataset generation process. We summarize the key steps of our VTKG construction process as follows.
>  - Step 1. To assemble visually expressible commonsense knowledge, we collected visual phrases from four computer vision benchmark datasets: VRD, UnRel, HICO-DET, and VisKE, where the first three datasets also provide images (Lines 158-180).
>  - Step 2. We constructed ConceptNetW from ConceptNet by converting the triplets into a form consistent with the other datasets (Lines 196-208).
>  - Step 3. We constructed WN18RR++ from WN18RR by mapping each entity into a unique synset ID (Lines 209-217).
>  - Step 4. We constructed VTKG-I by combining VRD, UnRel, and HICO-DET. Also, we constructed VTKG-C by combining all the above datasets (Table 1). When we combined the datasets, we replaced each entity/relation in a triplet with a synset ID defined in WordNet to express triplets using the vocabulary defined in WordNet. All triplets (except those from WN18RR++ since WN18RR++ is a subset of WordNet) in our VTKGs were inspected and validated by human annotators (Lines 218-248 & Lines 812-826).
>
> $\rule{18cm}{0.05cm}$
>
> > The quality of VTKG is unclear. In this paper, rather than building a new multi-modal knowledge graph from scratch, the authors design a pipeline to extract and merge triplets from CV and CSKG datasets. This kind of method may cause error accumulation in each step of the pipeline and thus reducing the overall quality of the final knowledge graph. Additionally, both evaluation of VTKG (E.g. matching degree of triplet and its image) and validation of each step in their pipeline are missing in this paper. All these things above make me worried about the quality of their constructed VTKG.
>
> ## Validations of VTKGs
> When creating VTKGs, we utilized the existing benchmark computer vision datasets and well-known knowledge bases, which are already widely used in research. We manually inspected and validated all triplets (except those from WN18RR++ since WN18RR++ is a subset of a standard lexical database, WordNet) in our VTKGs, as described in **Appendix A**. We developed the data labeling tool shown in Figure 6 to inspect each triplet to determine whether the triplet is valid and map each entity/relation in a triplet to an appropriate synset ID in WordNet (Lines 812-826). During this process, 269 triplets were removed among 19,065 original triplets; we removed invalid or hard-to-interpret triplets, e.g., $\langle \texttt{slave}, \texttt{hang\\_from}, \texttt{st} \rangle$. When sampling 2,000 triplets and double-checking the labels, the labeling accuracy was 95.01%.
>
> ## Additional Validations of VTKGs
> Additionally, we checked the matching of each triplet and its image. In our datasets, 1,316 triplets are expressed by images, and there are 141,247 images for triplets. The triplet images came from VRD, UnRel, and HICO-DET, well-known computer vision benchmarks. To examine whether a triplet is well matched with its image, we randomly sampled 100 images for each triplet. If a triplet has less than 100 images, we examined all the images for the triplet. As a result, we examined 36,919 images, and each image was examined by two different annotators. We consider a triplet is matched with its image only if both annotators agree on it, i.e., if one of the annotators disagrees, we consider the triplet is not matched with its image. When measuring the matching degree in this way, the accuracy was 98.62%. In a revised paper, we will describe this process of validating the matching between triplets and their images.
>
> ## Examples of Triplets in VTKGs
> We provide some examples of the triplets and their images and descriptions in our VTKG datasets. As explained above, each entity or relation is represented using a synset of WordNet, e.g., ‘airplane.n.01’ or ‘have.v.01’, where ‘n’ indicates a noun, ‘v’ indicates a verb, and the last two digits indicate the sense number. The descriptions of entities and relations are from WordNet. Since we cannot use an external link here, we provide the image IDs of the source dataset for the triplet images. If our paper is accepted, we will release the full datasets and codes.
>
> ____
>
> ### **(airplane.n.01, have.v.01, wheel.n.01)**
>
> #### **[Descriptions]**
>
> **airplane.n.01**: an aircraft that has a fixed wing and is powered by propellers or jets\
> **have.v.01**: have or possess, either in a concrete or an abstract sense\
> **wheel.n.01**: a simple machine consisting of a circular frame with spokes (or a solid disc) that can rotate on a shaft or axle (as in vehicles or other machines)
>
> #### **[Images]**
>
> 5406472568_78d687217f_b.jpg, 6994124346_d4813d9602_b.jpg, 8059265370_6843fc7ba4_b.jpg from **VRD**
>
> ____
>
> ### **(kite.n.03, fly.v.01, sky.n.01)**
>
> #### **[Descriptions]**
>
> **kite.n.03**: plaything consisting of a light frame covered with tissue paper; flown in wind at end of a string\
> **fly.v.01**: travel through the air; be airborne\
> **sky.n.01**: the atmosphere and outer space as viewed from the earth
>
> #### **[Images]**
>
> 4433556373_ddfabedff5_o.jpg, 8642248228_d2187daf86_b.jpg, 9759509291_143489b32a_b.jpg from **VRD**
>
> ____
>
> ### **(person.n.01, drive.v.01, car.n.01)**
>
> #### **[Descriptions]**
>
> **person.n.01**: a human being\
> **drive.v.01**: operate or control a vehicle\
> **car.n.01**: a motor vehicle with four wheels; usually propelled by an internal combustion engine
>
> #### **[Images]**
>
> HICO_train2015_00002944.jpg, HICO_test2015_00005622.jpg from **HICO-DET**\
> 3141015389_d1dd4fa2ee_b.jpg from **VRD**
>
> ____
>
> ## More Statistics of VTKGs
> The following tables, Table A1 and Table A2 show some additional statistics about our VTKG datasets, which will be used for interpreting the results of our ablation studies shown below.
>
> ### **Table A1: Entities and relations with and without visual features**
>
> $\begin{array}{rcc}
> \hline
>  & \text{VTKG-I} & \text{VTKG-C} \\\\
> \hline
> \text{Entities w/ $\bf{h}\_\text{vis}$} & 181\ (100\\%) & 7,863\ (18.2\\%) \\\\
> \text{Entities w/o $\bf{h}\_\text{vis}$} & 0\ (0\\%) & 35,404\ (81.8\\%) \\\\
> \text{Total} & 181 & 43,267 \\\\
> \hline
> \text{Relations w/ $\\mathbf{r}\_\\text{vis}$} & 217\ (100\\%) & 217\ (7.9\\%) \\\\
> \text{Relations w/o $\\mathbf{r}\_\\text{vis}$} & 0\ (0\\%) & 2,514\ (92.1\\%) \\\\
> \text{Total} & 217 & 2,731 \\\\
> \hline
> \end{array}$
>
> ### **Table A2: Categorizing triplets based on whether their entities and relations have visual features**
>
> $\begin{array}{rcc}
> \hline
>  & \text{VTKG-I} & \text{VTKG-C} \\\\
> \hline
> \text{two entities w/ visual features \\& relation w/ visual features} & 1,316\ (100\\%) & 2,569\ (2.3\\%) \\\\
> \text{one entity w/ visual features \\& relation w/ visual features} & 0\ (0\\%) & 1,743\ (1.6\\%) \\\\
> \text{two entities w/o visual features \\& relation w/ visual features} & 0\ (0\\%) & 522\ (0.5\\%) \\\\
> \hline
> \text{two entities w/ visual features \\& relation w/o visual features} & 0\ (0\\%) & 10,166\ (9.1\\%) \\\\
> \text{one entity w/ visual features \\& relation w/o visual features} & 0\ (0\\%) & 21,106\ (18.9\\%) \\\\
> \text{two entities w/o visual features \\& relation w/o visual features} & 0\ (0\\%) & 75,385\ (67.6\\%) \\\\
> \hline
> \text{Total} & 1,316 & 111,491 \\\\
> \hline
> \end{array}$
>
> $\rule{18cm}{0.05cm}$
>
> > The discussion of VISTA’s each component in section 5.4 is too brief, considering VISTA is quite complex with many different components. Conclusions and analysis like which component play a more critical role in VISTA should be concluded.
>
> ## More Experiments for Ablation Studies
>
> We added more ablation studies in Table A3. The blue colored rows indicate the new experiments. The leftmost column indicates a higher-level category of our ablation study. Note that Table A3 also includes all results shown in Table 5 for complete analysis.
>  - Modality: We show the results of VISTA using different combinations of the modalities. Recall that $\bf{h}\_\text{vis}$ indicates the visual feature of an entity, $\bf{h}\_\text{txt}$ indicates the textual feature of an entity, $\bf{r}\_\text{vis}$ indicates the visual feature of a relation, and $\bf{r}\_\text{txt}$ indicates the textual feature of a relation.
>  - Entity Modality: We show the impact of different modalities for entities.
>  - Relation Modality: We show the impact of different modalities for relations.
>  - Encoder: We added the ablation study of removing the entity transformer in VISTA, where we replace the entity transformer with a simple learnable vector $\bf{h}$ for each entity.
>  - Decoder: We copied the results regarding the decoder from Table 5 for complete comparisons.
>
> ### **Table A3: Additional ablation studies**
>
> $\begin{array}{|c|c|c|c|}\hline&&\text{VTKG-I}&\text{VTKG-C} \\\\&&\text{MR\hspace{23mu}MRR\hspace{23mu}Hit@1\hspace{18mu}Hit@3\hspace{15mu}Hit@10} & \text{\hspace{-2.5mu}MR\hspace{30.5mu}MRR\hspace{24mu}Hit@1\hspace{18mu}Hit@3\hspace{15mu}Hit@10\hspace{-5mu}}
> \\\\
> \hline
> \begin{array}{c} \text{Modality} \\\\ \text{(Entity + Relation)} \end{array} & \begin{array}{c}\text{\textcolor{blue}{w/o $\bf{h}\_\text{vis}$, $\bf{h}\_\text{txt}$, $\bf{r}\_\text{vis}$, $\bf{r}\_\text{txt}$}} \\\\ \text{\textcolor{blue}{w/o $\bf{h}\_\text{vis}$, $\bf{r}\_\text{vis}$}} \\\\ \text{\textcolor{blue}{w/o $\bf{h}\_\text{txt}$, $\bf{r}\_\text{txt}$}} \end{array} & \hspace{6mu}\begin{array}{ccccc} 23.3 & 0.3979 & 0.3092 &0.4313 & 0.5725 \\\\ 21.0& 0.4423& 0.3473& 0.4542& 0.6450 \\\\ 19.5& 0.4525& 0.3626& 0.4733& 0.6336 \end{array}\hspace{10mu} & \begin{array}{ccccc} 3163.9 &0.3651 & 0.3252 & 0.3806 & 0.4406 \\\\  238.7& 0.4556& 0.3827& 0.4811& 0.6014 \\\\ 2641.1& 0.3743& 0.3331& 0.3893& 0.4523 \end{array}\hspace{10mu}\\\\
> \hline
> \text{Entity Modality}& \begin{array}{c}\text{\textcolor{blue}{w/o $\bf{h}\_\text{vis}$, $\bf{h}\_\text{txt}$}} \\\\ \text{\textcolor{blue}{w/o $\bf{h}\_\text{vis}$}} \\\\ \text{\textcolor{blue}{w/o $\bf{h}\_\text{txt}$}} \end{array} &\hspace{6mu}\begin{array}{ccccc} 22.2& 0.3970& 0.2939& 0.4389& 0.5916 \\\\ 20.5& 0.4149& 0.3053& 0.4542& 0.6221 \\\\ 17.2& 0.4590& 0.3588& 0.4962& 0.6641 \end{array}\hspace{10mu} & \begin{array}{ccccc} 3036.6& 0.3661& 0.3268& 0.3816& 0.4403 \\\\ 229.1& 0.4569& 0.3832& 0.4842& 0.6028 \\\\ 2599.5& 0.3721& 0.3312& 0.3871& 0.4490 \end{array}\hspace{10mu}\\\\
> \hline
> \text{Relation Modality}& \begin{array}{c} \text{\textcolor{blue}{w/o $\bf{r}\_\text{vis}$, $\bf{r}\_\text{txt}$}} \\\\ \text{\textcolor{blue}{w/o $\bf{r}\_\text{vis}$}} \\\\ \text{\textcolor{blue}{w/o $\bf{r}\_\text{txt}$}} \\\\ \text{replace $\bf{r_{\text{vis}}}$ w/ $\bf{u}$} \end{array} &\hspace{6mu}\begin{array}{ccccc} 18.8& 0.4297& 0.3244& 0.4656& 0.6412 \\\\ 19.0& 0.4546& 0.3588& 0.4924& 0.6412 \\\\ 17.3& 0.4587& 0.3511& 0.5000& 0.6641 \\\\ 18.3& 0.4334& 0.3359& 0.4580& 0.6489 \end{array}\hspace{10mu} & \begin{array}{ccccc} 235.8& 0.4601& 0.3856& 0.4865& 0.6097 \\\\ 223.7& 0.4661& 0.3910& 0.4935& 0.6127 \\\\ 236.3& 0.4608& 0.3845& 0.4903& 0.6088 \\\\ \hspace{4.5mu}318.1\hspace{4.5mu}& 0.4159& 0.3415& 0.4454& 0.5573 \end{array}\hspace{10mu}\\\\
> \hline
> \text{Encoder}& \begin{array}{c} \text{w/o $\bf{z_\text{\[ENT\]}}$ and $\bf{z_\text{\[REL\]}}$}\\\\ \text{w/o $\bf{h}$ and $\bf{r}$} \\\\ \text{w/o rel. trans.} \\\\ \text{\textcolor{blue}{w/o ent. trans.}} \end{array} &\hspace{6mu}\begin{array}{ccccc} 18.6&0.4022&0.2901&0.4542&0.6221 \\\\ 19.0&0.4376&0.3282&0.4847&0.6489 \\\\ 18.4&0.4168&0.3015&0.4733&0.6221 \\\\ 25.2&0.4065&0.3321&0.4237&0.5458  \end{array}\hspace{10mu} & \begin{array}{ccccc}\hspace{4.5mu} 316.8 \hspace{4.5mu}& 0.4018& 0.3251& 0.4340& 0.5486 \\\\ 250.9&0.4497&0.3719&0.4835&0.5976 \\\\ 369.3&0.3889&0.3119&0.4203&0.5381 \\\\ 4368.7&0.3615&0.3194&0.3775&0.4442 \end{array}\hspace{10mu}\\\\
> \hline
> \text{Decoder} & \begin{array}{c} \text{w/ DistMult decoder} \\\\ \text{replace dot prod. w/ lin.} \end{array} &\hspace{6mu}\begin{array}{ccccc} 21.0& 0.3911& 0.2824& 0.4160& 0.6336 \\\\ 29.5& 0.4063& 0.3168& 0.4351& 0.5802 \end{array}\hspace{10mu} & \begin{array}{ccccc} \hspace{4.5mu}420.6\hspace{4.5mu}& 0.4195& 0.3572& 0.4385& 0.5419 \\\\ 964.5& 0.4297& 0.3796& 0.4478& 0.5236 \end{array}\hspace{10mu}\\\\
> \hline
> &\text{VISTA} & \hspace{6mu}\begin{array}{ccccc} 17.3& 0.4650& 0.3626& 0.5076& 0.6641 \end{array}\hspace{10mu} & \begin{array}{ccccc} \hspace{4.5mu}220.8\hspace{4.5mu}& 0.4675& 0.3918& 0.4961& 0.6157 \end{array}\hspace{10mu}\\\\
> \hline
> \end{array}$
>
> ## Conclusions and Analysis of the Additional Ablation Studies
> We summarize our conclusions drawn from our ablation studies as follows; we will add the results of our additional experiments and analysis in a revised paper.
>  - On both VTKG-I and VTKG-C, we see that adding visual or textual features leads to better performance than the case without those features. More importantly, using all modalities presented in the last row of Table A3 leads to the best performance.
>  - The impact of the visual features is more prominent on VTKG-I than on VTKG-C. To understand this point more thoroughly, we focus on the statistics about categorizing triplets based on whether their entities and relations have visual features, shown in Table A2. While all entities and relations have their visual features on VTKG-I, many entities and relations do not have visual features on VTKG-C. As a result, visual features are more important than textual features on VTKG-I, while textual features are more important than visual features on VTKG-C.
>  - When we compare the impact of the entity modality and the relation modality, removing a modality of entities is more critical than removing a modality of relations on both VTKG-I and VTKG-C.
>  - The performance decreases of VISTA with ‘replace $\bf{r}\_\text{vis}$ w/ $\bf{u}$’ indicate that our way of constructing $\bf{r}\_\text{vis}$ is more effective than just using the union bounding box of a triplet when creating a visual representation vector of $r$.
>  - By looking at the first two rows of Encoder in Table A3, we see that $\bf{z_\text{[ENT]}}$, $\bf{z_\text{[REL]}}$, $\bf{h}$, and $\bf{r}$ are all important inputs of the entity and relation encoding transformers. By looking at the last two rows of Encoder in Table A3, we note that removing either the entity encoding transformer or the relation encoding transformer greatly impacts VISTA's performance. Regarding Hit@1, removing the relation encoding transformer leads to a more critical decrease in performance than removing the entity encoding transformer. On the other hand, for the other metrics (i.e., MR, MRR, Hit@3, and Hit@10), the entity encoding transformer is more critical than the relation encoding transformer.
>  - The two rows of Decoder in Table A3 indicate that making changes in the way of predicting a missing entity for knowledge graph completion leads to degrading the performance of VISTA (Lines 517-520). This validates the effectiveness of our triplet decoding transformer.
>
> $\rule{18cm}{0.05cm}$
>
> > They do not clearly validate the effectiveness of introducing image information of triplet into the multimodal knowledge graph. One of the most important contributions they make in this paper is proposing to express triplet in knowledge graph with its image. However, they fail to quantitatively validate the effectiveness of doing so in the experiment section. I suggest adding a new ablation study to train each model w/ and w/o visual information of triplet.
>
> We present the results by removing triplet images $\mathcal{I}\_\{tri}$ in Table A4, where we also present the ratios of the affected triplets by removing $\mathcal{I}\_\{tri}$ (denoted by **'Aff.'**). Note that the numbers presented in the Aff. column are computed based on Table A2. The performance drops in VTKG-I confirm the effectiveness of triplet images. Even though the performance drops in VTKG-C seem not as significant as in VTKG-I, that is mainly because the ratios of the affected triplets (Aff.) by removing the visual information of triplets are not substantial in VTKG-C.
>
> ### **Table A4: Effect of triplet images**
>
> $\begin{array}{|c|c|c|}\hline&\text{VTKG-I}&\text{VTKG-C} \\\\
> &\text{Aff.\hspace{24mu}MR\hspace{23mu}MRR\hspace{21mu}Hit@10} & \text{\hspace{-1mu}Aff.\hspace{30mu}MR\hspace{30mu}MRR\hspace{21mu}Hit@10}
> \\\\
> \hline
> \begin{array}{c}
> \text{w/o $\mathcal{I}\_\text{tri}$}
> \end{array}
> & \begin{array}{cccc}
> 100\\% & \hspace{1.5mu}19.0\hspace{1.5mu} & \hspace{1.5mu}0.4546\hspace{1.5mu} & 0.6412
> \end{array}\hspace{10mu}
> & \begin{array}{cccc}
> \hspace{3mu}4.3\\% & \hspace{6mu}223.7 & \hspace{3mu}0.4661\hspace{3mu} & 0.6127
> \end{array}\hspace{10mu}\\\\
> \text{VISTA}
> &\hspace{6mu}
> \begin{array}{ccccc}
> \hspace{3mu}\text{N/A}& \hspace{1mu}\bf{17.3} & \hspace{-3mu}\bf{0.4650} & \hspace{-5mu}\bf{0.6641}
> \end{array}
> \hspace{10mu}
> &\begin{array}{ccccc}
> \hspace{7.5mu}\text{N/A} & \hspace{4mu}\bf{220.8}& \hspace{-2mu}\bf{0.4675} & \hspace{-5mu}\bf{0.6157}
> \end{array}
> \hspace{10mu}\\\\
> \hline
> \end{array}$
>
> $\rule{18cm}{0.05cm}$
>
> > They could also consider training models in VTKG and another existing multi-modal knowledge graph respectively, and then testing in a new knowledge graph dataset (out-of-domain setting).
>
> Throughout the paper, we assume the conventional transductive setting for knowledge graph completion. That is, all entities and relations are assumed to be observed during training, and what we predict during testing is a plausible combination of the entities and relations. This is the setting that most existing multi-modal knowledge graph embedding methods assume. To make predictions in the out-of-domain setting, we should consider an inductive learning scenario, which is out of the scope of our paper; this can be an interesting future work.

---

### Official Review · Reviewer_jTyB · 2023-08-12

**Soundness:** 3

**Excitement:**

3: Ambivalent: It has merits (e.g., it reports state-of-the-art results, the idea is nice), but there are key weaknesses (e.g., it describes incremental work), and it can significantly benefit from another round of revision. However, I won't object to accepting it if my co-reviewers champion it.

**Paper Topic And Main Contributions:**

The research question of the paper revolves around enhancing knowledge graphs by incorporating visual and textual descriptions, specifically through the concept of visual-textual knowledge graphs (VTKGs). The authors introduce a novel type of knowledge graph representation VTKGs where not only entities, but also triplets, can be represented using images. They propose a method called VISTA for learning the representation of VTKGs, utilizing transformers for entity and relation encoding as well as triplet decoding. The main contributions of the paper include the introduction of VTKGs, the creation of benchmark datasets with visually explainable triplets, and the development of the VISTA method, which outperforms existing knowledge graph completion methods when applied to real-world VTKGs.

**Reasons To Accept:**

* The introduction of visual-textual knowledge graphs (VTKGs) marks an innovative step in knowledge graph representation. It goes beyond conventional approaches by incorporating both images and text descriptions for entities and triplets.
* The creation of benchmark datasets with visually explainable triplets is an important contribution which could interest the researchers in the KG realm.
* The development of the VISTA method, encompassing entity encoding, relation encoding, and triplet decoding transformers, showcases a sophisticated approach to learning knowledge graph representations in VTKGs.
* They conducted extensive and comprehenisve experiments on four datasets with 10 different baselines. Experimental results revealed the real-world efficacy of the VISTA method by outperforming all baseline methods of knowledge graph completion.

**Reasons To Reject:**

* The writing of the paper is not always clear, especially the description in the beginning part, making it not easy to understand the overview of the method.
* Maybe you can use a more illustrative Figure 1 to elaborate the difference of the KG representation discussed in this paper from previous multimodal representations.


**Reproducibility:**

4: Could mostly reproduce the results, but there may be some variation because of sample variance or minor variations in their interpretation of the protocol or method.

**Reviewer Confidence:**

2: Willing to defend my evaluation, but it is fairly likely that I missed some details, didn't understand some central points, or can't be sure about the novelty of the work.

---

> ### Author Rebuttal · Authors · 2023-08-28
>
> We appreciate your time and effort in reading our paper. In what follows, we provide our answers to your concerns. Please also consider that we prepared our answers to **seven different reviewers** during the rebuttal period. By incorporating the comments from seven reviewers, including yours, we got a chance to improve our paper. We hope the improvements can be considered in the final assessment of our paper.
>
> $\rule{18cm}{0.05cm}$
>
> > The writing of the paper is not always clear, especially the description in the beginning part, making it not easy to understand the overview of the method.
>
> We will refine the writing and more clearly describe the overview of our method in a revised paper. Any specific suggestions would be appreciated.
>
> $\rule{18cm}{0.05cm}$
>
> > Maybe you can use a more illustrative Figure 1 to elaborate the difference of the KG representation discussed in this paper from previous multimodal representations.
>
> Thank you for the suggestion! We will add a direct comparison between our VTKG and an existing multimodal knowledge graph (MKG) in Figure 1. The main similarities and differences between our VTKGs and existing MKGs are as follows.
>  - Both can have images and text descriptions for entities.
>  - While VTKGs can have images for triplets, MKGs do not assume that triplets can be represented using images. For example, in Figure 1, $\langle \texttt{person}, \texttt{pull}, \texttt{cart}\rangle$ and $\langle \texttt{person}, \texttt{ride}, \texttt{horse}\rangle$ are represented by images in our VTKG. These triplet images are utilized only in VTKGs but not in MKGs.
>  - While VTKGs can have text descriptions of relations, most MKGs do not consider describing the lexical meanings of relations using text.

---

### Official Review · Reviewer_K6Ph · 2023-08-12

**Paper Topic And Main Contributions:** 1. This paper proposes a visual-textu…
**Soundness:** 3

**Excitement:**

3: Ambivalent: It has merits (e.g., it reports state-of-the-art results, the idea is nice), but there are key weaknesses (e.g., it describes incremental work), and it can significantly benefit from another round of revision. However, I won't object to accepting it if my co-reviewers champion it.

**Reasons To Accept:**

1. Detailed experiment and full ablation studies
2. Clear and well-written

**Reasons To Reject:**

1. The results of the baselines deviate from what was presented in the original paper.

2. The benchmark is collected from other datasets and lacks more analysis and validation.

3. VISTA appears to be a mere combination of existing methods, and less innovative. Like Entity Encoding and Relation Encoding Transformer.

4. The overall structure seems heavy and not easily scalable.

**Reproducibility:**

3: Could reproduce the results with some difficulty. The settings of parameters are underspecified or subjectively determined; the training/evaluation data are not widely available.

**Reviewer Confidence:**

3: Pretty sure, but there's a chance I missed something. Although I have a good feel for this area in general, I did not carefully check the paper's details, e.g., the math, experimental design, or novelty.

---

> ### Author Rebuttal · Authors · 2023-08-28
>
> We appreciate your time and effort in reading our paper. In what follows, we provide our answers to your concerns. Please also consider that we prepared our answers to **seven different reviewers** during the rebuttal period. By incorporating the comments from seven reviewers, including yours, we got a chance to improve our paper. We hope the improvements can be considered in the final assessment of our paper.
>
> $\rule{18cm}{0.05cm}$
> > 1. The results of the baselines deviate from what was presented in the original paper.
>
> As described in Lines 424-427, we reproduced the results of the baselines by setting the dimension to be $d=256$ for all methods for a fair comparison; in MKGformer, $d$ is fixed to 768 since it directly uses ViT and BERT. Details about how we run the baseline methods are described in Appendix B. Among the datasets, FB15K237 is the only dataset on which we can check the baselines’ performances in their original papers because the other datasets were introduced in our paper. Among the baselines, ComplEx-N3, RotatE, PairRE, MKGformer, MoSE, and IMF reported their results on FB15K237; the other baselines, ANALOGY, RSME, TransAE, and OTKGE did not use FB15K237 in their papers. In Table A1, we present the baselines’ performances on FB15K237 reported in their original papers and the reproduced results; ‘N/A’ indicates the case where the original paper does not provide the result. We run VISTA with $d=256$ (reported in our original submission) and $d=512$ (added results during the rebuttal period) for comparisons with the baseline methods.
>
> ### **Table A1: Performances of baseline methods and VISTA on FB15K237**
>
> $\begin{array}{|c|c|c|}
> \hline
> &&\begin{array}{c}
> \text{FB15K237}\\\\
> \text{\hspace{-12mu}MR\hspace{33mu}MRR\hspace{31mu}Hit@1\hspace{26mu}Hit@3\hspace{22mu}Hit@10\hspace{-15mu}}
> \end{array}\\\\
> \hline
> \text{ComplEx-N3}
> &\begin{array}{c}
> \text{Original Paper ($d=4000$)}\\\\
> \text{Reproduced Results ($d=4000$)}\\\\
> \text{Reproduced Results ($d=256$)}
> \end{array}
> &\begin{array}{ccccc}
> \hspace{-1mu}\text{N/A}\hspace{7mu}&0.3700\hspace{7.5mu}&\text{N/A}\hspace{7.5mu}&\text{N/A}\hspace{7.5mu}&0.5600\hspace{4mu}\\\\
> \hspace{-1mu}143.5\hspace{7mu}&0.3666\hspace{7.5mu}&0.2713\hspace{7.5mu}&0.4018\hspace{7.5mu}&0.5587\hspace{4mu}\\\\
> \hspace{-1mu}172.7\hspace{7mu}&0.3510\hspace{7.5mu}&0.2584\hspace{7.5mu}&0.3847\hspace{7.5mu}&0.5391\hspace{4mu}
> \end{array}\\\\
> \hline
> \text{RotatE}
> &\begin{array}{c}
> \text{Original Paper ($d=2000$)}\\\\
> \text{Reproduced Results ($d=256$)}
> \end{array}
> &\begin{array}{ccccc}
> \hspace{-1mu}177.0\hspace{7mu}&0.3380\hspace{7.5mu}&0.2410\hspace{7.5mu}&0.3750\hspace{7.5mu}&0.5330\hspace{4mu}\\\\
> \hspace{-1mu}246.1\hspace{7mu}&0.3099\hspace{7.5mu}&0.2183\hspace{7.5mu}&0.3433\hspace{7.5mu}&0.4932\hspace{4mu}
> \end{array}\\\\
> \hline
> \text{PairRE}
> &\begin{array}{c}
> \text{Original Paper ($d=2000$)}\\\\
> \text{Reproduced Results ($d=256$)}
> \end{array}
> &\begin{array}{ccccc}
> \hspace{-1mu}160.0\hspace{7mu}&0.3510\hspace{7.5mu}&0.2560\hspace{7.5mu}&0.3870\hspace{7.5mu}&0.5440\hspace{4mu}\\\\
> \hspace{-1mu}184.3\hspace{7mu}&0.3326\hspace{7.5mu}&0.2399\hspace{7.5mu}&0.3675\hspace{7.5mu}&0.5193\hspace{4mu}
> \end{array}\\\\
> \hline
> \text{MKGformer}
> &\begin{array}{c}
> \text{Original Paper ($d=768$)}\\\\
> \text{Reproduced Results ($d=768$)}
> \end{array}
> &\begin{array}{ccccc}
> \hspace{-1mu}221.0\hspace{7mu}&\text{N/A}\hspace{7.5mu}&0.2560\hspace{7.5mu}&0.3670\hspace{7.5mu}&0.5040\hspace{4mu}\\\\
> \hspace{-1mu}297.6\hspace{7mu}&0.3095\hspace{7.5mu}&0.2278\hspace{7.5mu}&0.3356\hspace{7.5mu}&0.4740\hspace{4mu}
> \end{array}\\\\
> \hline
> \text{MoSE-AI}
> &\begin{array}{c}
> \text{Original Paper ($d=4000$)}\\\\
> \text{Reproduced Results ($d=4000$)}\\\\
> \text{Reproduced Results ($d=256$)}
> \end{array}
> &\begin{array}{ccccc}
> \hspace{-1mu}135.0\hspace{7mu}&\text{N/A}\hspace{7.5mu}&0.2550\hspace{7.5mu}&0.3760\hspace{7.5mu}&0.5180\hspace{4mu}\\\\
> \hspace{-1mu}134.9\hspace{7mu}&0.3397\hspace{7.5mu}&0.2514\hspace{7.5mu}&0.3704\hspace{7.5mu}&0.5144\hspace{4mu}\\\\
> \hspace{-1mu}149.1\hspace{7mu}&0.3247\hspace{7.5mu}&0.2384\hspace{7.5mu}&0.3532\hspace{7.5mu}&0.4965\hspace{4mu}
> \end{array}\\\\
> \hline
> \text{MoSE-BI}
> &\begin{array}{c}
> \text{Original Paper ($d=4000$)}\\\\
> \text{Reproduced Results ($d=4000$)}\\\\
> \text{Reproduced Results ($d=256$)}
> \end{array}
> &\begin{array}{ccccc}
> \hspace{-1mu}117.0\hspace{7mu}&\text{N/A}\hspace{7.5mu}&0.2810\hspace{7.5mu}&0.4110\hspace{7.5mu}&0.5650\hspace{4mu}\\\\
> \hspace{-1mu}117.8\hspace{7mu}&0.3735\hspace{7.5mu}&0.2792\hspace{7.5mu}&0.4091\hspace{7.5mu}&0.5635\hspace{4mu}\\\\
> \hspace{-1mu}132.3\hspace{7mu}&0.3466\hspace{7.5mu}&0.2570\hspace{7.5mu}&0.3755\hspace{7.5mu}&0.5303\hspace{4mu}
> \end{array}\\\\
> \hline
> \text{MoSE-MI}
> &\begin{array}{c}
> \text{Original Paper ($d=4000$)}\\\\
> \text{Reproduced Results ($d=4000$)}\\\\
> \text{Reproduced Results ($d=256$)}
> \end{array}
> &\begin{array}{ccccc}
> \hspace{-1mu}127.0\hspace{7mu}&\text{N/A}\hspace{7.5mu}&0.2680\hspace{7.5mu}&0.3940\hspace{7.5mu}&0.5400\hspace{4mu}\\\\
> \hspace{-1mu}140.9\hspace{7mu}&0.3513\hspace{7.5mu}&0.2613\hspace{7.5mu}&0.3854\hspace{7.5mu}&0.5298\hspace{4mu}\\\\
> \hspace{-1mu}148.9\hspace{7mu}&0.3275\hspace{7.5mu}&0.2416\hspace{7.5mu}&0.3568\hspace{7.5mu}&0.4984\hspace{4mu}
> \end{array}\\\\
> \hline
> \text{IMF}
> &\begin{array}{c}
> \text{Original Paper ($d=256$)}\\\\
> \text{Reproduced Results ($d=256$)}
> \end{array}
> &\begin{array}{ccccc}
> \hspace{-1mu}134.0\hspace{7mu}&0.3890\hspace{7.5mu}&0.2870\hspace{7.5mu}&\text{N/A}\hspace{7.5mu}&0.5930\hspace{4mu}\\\\
> \hspace{-1mu}151.8\hspace{7mu}&0.3677\hspace{7.5mu}&0.2735\hspace{7.5mu}&0.4040\hspace{7.5mu}&0.5573\hspace{4mu}
> \end{array}\\\\
> \hline
> \text{VISTA}
> &\begin{array}{c}
> \text{Original Submission ($d=256$)}\\\\
> \text{Added Results ($d=512$)}
> \end{array}
> &\begin{array}{ccccc}
> 122.3&0.3732&0.2796&0.4081&0.5633\hspace{4mu}\\\\
> \bf{115.3}&\bf{0.3788}&\bf{0.2854}&\bf{0.4144}&\bf{0.5670}\hspace{4mu}
> \end{array}\\\\
> \hline
> \end{array}$
>
> In Table A1, the key things to note are as follows:
>  - The model performances are affected by the dimension ($d$); thus, using the same dimension for all methods is needed for a fair comparison, as we did in our paper.
>  - VISTA outperforms the baseline methods even when it uses a smaller dimension than the baseline methods.
>  - The authors of MKGformer[1] reported that their original implementation had a severe bug leading to data leakage problems. After fixing the bug, the performance of MKGformer was degraded. In Table A1, we use the fixed results reported on arXiv[2]. When we ran MKGformer, we had to decrease the batch size (from 96 to 32) due to the memory constraint of our GPUs.
>  - The results of IMF are not precisely reproducible.
>
> **References:**\
> [1] X. Chen et al., Hybrid Transformer with Multi-level Fusion for Multimodal Knowledge Graph Completion, SIGIR 2022\
> [2] X. Chen et al., Hybrid Transformer with Multi-level Fusion for Multimodal Knowledge Graph Completion, arXiv:2205.02357v4
>
> $\rule{18cm}{0.05cm}$
>
> > 2. The benchmark is collected from other datasets and lacks more analysis and validation.
>
> ## Validations of VTKGs
> When creating VTKGs, we utilized the existing benchmark computer vision datasets and well-known knowledge bases, which are already widely used in research. We manually inspected and validated all triplets (except those from WN18RR++ since WN18RR++ is a subset of a standard lexical database, WordNet) in our VTKGs, as described in **Appendix A**. We developed the data labeling tool shown in Figure 6 to inspect each triplet to determine whether the triplet is valid and map each entity/relation in a triplet to an appropriate synset ID in WordNet (Lines 812-826). During this process, 269 triplets were removed among 19,065 original triplets; we removed invalid or hard-to-interpret triplets, e.g., $\langle \texttt{slave}, \texttt{hang\\_from}, \texttt{st} \rangle$. When sampling 2,000 triplets and double-checking the labels, the labeling accuracy was 95.01%.
>
> ## Additional Validations of VTKGs
> Additionally, we checked the matching of each triplet and its image. In our datasets, 1,316 triplets are expressed by images, and there are 141,247 images for triplets. The triplet images came from VRD, UnRel, and HICO-DET, well-known computer vision benchmarks. To examine whether a triplet is well matched with its image, we randomly sampled 100 images for each triplet. If a triplet has less than 100 images, we examined all the images for the triplet. As a result, we examined 36,919 images, and each image was examined by two different annotators. We consider a triplet is matched with its image only if both annotators agree on it, i.e., if one of the annotators disagrees, we consider the triplet is not matched with its image. When measuring the matching degree in this way, the accuracy was 98.62%. In a revised paper, we will describe this process of validating the matching between triplets and their images.
>
> ## Examples of Triplets in VTKGs
> We provide some examples of the triplets and their images and descriptions in our VTKG datasets. As explained above, each entity or relation is represented using a synset of WordNet, e.g., ‘airplane.n.01’ or ‘have.v.01’, where ‘n’ indicates a noun, ‘v’ indicates a verb, and the last two digits indicate the sense number. The descriptions of entities and relations are from WordNet. Since we cannot use an external link here, we provide the image IDs of the source dataset for the triplet images. If our paper is accepted, we will release the full datasets and codes.
>
> --------------------------
> ### **(airplane.n.01, have.v.01, wheel.n.01)**
>
> #### **[Descriptions]**
>
> **airplane.n.01**: an aircraft that has a fixed wing and is powered by propellers or jets\
> **have.v.01**: have or possess, either in a concrete or an abstract sense\
> **wheel.n.01**: a simple machine consisting of a circular frame with spokes (or a solid disc) that can rotate on a shaft or axle (as in vehicles or other machines)
>
> #### **[Images]**
>
> 5406472568_78d687217f_b.jpg, 6994124346_d4813d9602_b.jpg, 8059265370_6843fc7ba4_b.jpg from **VRD**
>
> --------------------------
>
> ### **(kite.n.03, fly.v.01, sky.n.01)**
>
> #### **[Descriptions]**
>
> **kite.n.03**: plaything consisting of a light frame covered with tissue paper; flown in wind at end of a string\
> **fly.v.01**: travel through the air; be airborne\
> **sky.n.01**: the atmosphere and outer space as viewed from the earth
>
> #### **[Images]**
>
> 4433556373_ddfabedff5_o.jpg, 8642248228_d2187daf86_b.jpg, 9759509291_143489b32a_b.jpg from **VRD**
>
> --------------------------
>
> ### **(person.n.01, drive.v.01, car.n.01)**
>
> #### **[Descriptions]**
>
> **person.n.01**: a human being\
> **drive.v.01**: operate or control a vehicle\
> **car.n.01**: a motor vehicle with four wheels; usually propelled by an internal combustion engine
>
> #### **[Images]**
>
> HICO_train2015_00002944.jpg, HICO_test2015_00005622.jpg from **HICO-DET**\
> 3141015389_d1dd4fa2ee_b.jpg from **VRD**
>
> ____
>
> ## More Analysis of VTKGs
> The following tables show some additional statistics about our VTKG datasets.
> ### **Table A2: Entities and relations with and without visual features**
>
> $\begin{array}{rcc}
> \hline
>  & \text{VTKG-I} & \text{VTKG-C} \\\\
> \hline
> \text{Entities w/ $\bf{h}\_\text{vis}$} & 181\ (100\\%) & 7,863\ (18.2\\%) \\\\
> \text{Entities w/o $\bf{h}\_\text{vis}$} & 0\ (0\\%) & 35,404\ (81.8\\%) \\\\
> \text{Total} & 181 & 43,267 \\\\
> \hline
> \text{Relations w/ $\\mathbf{r}\_\\text{vis}$} & 217\ (100\\%) & 217\ (7.9\\%) \\\\
> \text{Relations w/o $\\mathbf{r}\_\\text{vis}$} & 0\ (0\\%) & 2,514\ (92.1\\%) \\\\
> \text{Total} & 217 & 2,731 \\\\
> \hline
> \end{array}$
>
> ### **Table A3: Categorizing triplets based on whether their entities and relations have visual features**
>
> $\begin{array}{rcc}
> \hline
>  & \text{VTKG-I} & \text{VTKG-C} \\\\
> \hline
> \text{two entities w/ visual features \\& relation w/ visual features} & 1,316\ (100\\%) & 2,569\ (2.3\\%) \\\\
> \text{one entity w/ visual features \\& relation w/ visual features} & 0\ (0\\%) & 1,743\ (1.6\\%) \\\\
> \text{two entities w/o visual features \\& relation w/ visual features} & 0\ (0\\%) & 522\ (0.5\\%) \\\\
> \hline
> \text{two entities w/ visual features \\& relation w/o visual features} & 0\ (0\\%) & 10,166\ (9.1\\%) \\\\
> \text{one entity w/ visual features \\& relation w/o visual features} & 0\ (0\\%) & 21,106\ (18.9\\%) \\\\
> \text{two entities w/o visual features \\& relation w/o visual features} & 0\ (0\\%) & 75,385\ (67.6\\%) \\\\
> \hline
> \text{Total} & 1,316 & 111,491 \\\\
> \hline
> \end{array}$
>
> From Table A2 and Table A3, we note that all entities and relations have their visual features on VTKG-I, whereas many entities and relations do not have visual features on VTKG-C. This analysis was beneficial to interpreting the results of our additional ablation studies requested by Reviewer bFL1 and Reviewer V1gZ.
>
> $\rule{18cm}{0.05cm}$
>
> > 3. VISTA appears to be a mere combination of existing methods, and less innovative. Like Entity Encoding and Relation Encoding Transformer.
>
> To the best of our knowledge, our entity and relation encoding transformers and the triplet decoding transformer are different from any existing method. Since VISTA is the first method that can handle images of triplets and visual/textual features of relations, the model architecture should be fundamentally distinguished from the existing methods. We would be pleased to discuss this point further when you provide specific references.
>
> $\rule{18cm}{0.05cm}$
>
> > 4. The overall structure seems heavy and not easily scalable.
>
> We first want to remind you that we already mentioned the scalability issue of VISTA in our Limitations Section (Lines 558-568). We present the runtimes of the baselines and VISTA on VTKG-C in the following figure. For a fair comparison, we compare the runtimes of methods that can handle different modalities. Even though making VISTA more scalable is one of our future works, we note that VISTA is the second best in terms of runtime and is much faster than MKGformer, OTKGE, and IMF.
>
> ### **Figure A1: Runtimes of multimodal methods on VTKG-C**
> TransAE$\hspace{30.3mu}$▒▒▒▒▒▒▒ (3 hours 40 minutes) \
> MKGformer$\hspace{4.2mu}$▒▒▒▒▒▒▒▒▒▒▒▒▒▒▒▒▒▒▒▒▒▒▒▒▒▒▒▒▒ (15 hours) \
> OTKGE$\hspace{38.6mu}$▒▒▒▒▒▒▒▒▒▒▒▒▒▒▒▒▒▒▒▒▒▒▒▒▒▒▒▒▒▒▒▒▒▒▒▒▒ ≈ ▒▒▒ (80 hours) \
> MoSE$\hspace{47.2mu}$▒▒ (40 minutes) \
> IMF$\hspace{59.8mu}$▒▒▒▒▒▒▒▒▒▒▒▒▒▒▒▒▒▒▒▒▒▒▒▒▒ (13 hours) \
> **VISTA**$\hspace{44.6mu}$▒▒▒▒▒▒ (3 hours) \
> -----------------|-----------|-----------|-----------|-----------|-----------|-----------|----- ≈ -----|----- \
> $\hspace{76.5mu}$0h$\hspace{43.0mu}$3h$\hspace{43.0mu}$6h$\hspace{43.0mu}$9h$\hspace{40.0mu}$12h$\hspace{34.5mu}$15h$\hspace{34.5mu}$18h$\hspace{45.5mu}$80h

---

### Meta-Review · Area_Chair_8nw8 · 2023-09-19

**Recommendation:** 3

**Metareview:**

In this paper, the authors propose a visual-textual knowledge graphs (VTKGs) benchmark datasets, where the triplet itself is explained using images and provides a detailed description of the meaning of entities and relationships. Meanwhile, they propose a knowledge graph representation learning method named VISTA that incorporates the visual and textual representations of entities and relations using entity relation encoding. The visual features are extracted by ViT-Base, and the textual features are extracted by BERT. Experimental results on four real-world datasets demonstrate that VISTA outperforms 10 different SOTA methods. As the reviewers mentioned, the paper writing should be improved and the authors should add the new results mentioned in rebuttal stage to revised version.

---

### Decision · Program_Chairs · 2023-10-07

**Decision:**

Accept-Findings

**Comment:**

In this paper, the authors propose a visual-textual knowledge graphs (VTKGs) benchmark datasets, where the triplet itself is explained using images and provides a detailed description of the meaning of entities and relationships. Meanwhile, they propose a knowledge graph representation learning method named VISTA that incorporates the visual and textual representations of entities and relations using entity relation encoding. The visual features are extracted by ViT-Base, and the textual features are extracted by BERT. Experimental results on four real-world datasets demonstrate that VISTA outperforms 10 different SOTA methods. As the reviewers mentioned, the paper writing should be improved and the authors should add the new results mentioned in rebuttal stage to revised version.